# Asymmetric Training with Heterogeneous Losses: A Probe into Architectural Resonance

## Abstract

Is deep learning generalization necessarily rooted in optimizing a single objective? We explore an alternative view: adaptive generalization may emerge from structured interactions among heterogeneous objectives. We propose an Asymmetric Training Paradigm that temporarily introduces non-competitive, per-class supervision (Sigmoid losses) into networks optimized with competitive softmax objectives. This is realized through orthogonally initialized auxiliary pathways, modulated by a scalar coefficient $\alpha$ and present only during training. Crucially, we employ strictly controlled experiments to rule out parameter count as a confounder, identifying that simple parameter expansion yields zero gain. Our mechanistic analysis reveals two effects: (1) The proposed topology (but not mere capacity) consistently smooths the initial optimization landscape. (2) Final performance exhibits an architecture-dependent pattern we term Architectural Resonance, where auxiliary signals benefit models only when aligned with inductive biases. A 6-block Vision Transformer (ViT-6L) exhibits constructive gradient alignment (cosine similarity $+0.19$), yielding absolute accuracy gains of $+9.2\%$ on CIFAR-100. By contrast, a CNN shows destructive conflicts (cosine similarity $-0.26$). We further corroborate this divergence in hybrid architectures (CoAtNet), highlighting a stage-dependent nature: transformer stages benefit from heterogeneity, while convolutional stages show limited compatibility. We validate scalability on ImageNet-1k, showing consistent top-1 gains for ViTs (up to $+2.25\%$ on ViT-B/16). Rather than functioning as a universal regularizer, our probe reveals that heterogeneous signals selectively benefit architectures with weak inductive biases (e.g., Vision Transformers), exposing a critical dependence between architectural flexibility and objective compatibility.

## 1 Introduction

A fundamental challenge in deep learning is understanding the complex interplay between a model's architectural inductive biases and the training strategies it is subjected to. While auxiliary supervision is a widely adopted technique for improving model performance (Szegedy et al., 2015; Lee et al., 2015; Caruana, 1997; Ruder, 2017), its application has been predominantly homogeneous, using objectives conceptually aligned with the main task. This raises a critical question we probe systematically: how do architectures intrinsically respond to fundamentally heterogeneous supervision? Specifically, how does a system designed for "winner-takes-all" competition (via softmax) react to signals that encourage "feature coexistence" (via sigmoid)?

To investigate this, we propose the Asymmetric Training Paradigm (Figure 1), a framework designed as a precise scientific probe. It temporarily introduces non-competitive, sigmoid-based supervision into a network through orthogonally initialized pathways, allowing us to systematically study the resulting internal dynamics. Our investigation reveals an architecture-dependent phenomenon that extends conventional understanding of auxiliary supervision, which we term Architectural Resonance. On CIFAR-100, this paradigm produces sharply divergent outcomes: Vision Transformers achieve an accuracy gain of $+9.2\%$, driven by constructive gradient synergy (cosine similarity $+0.19$), while Convolutional Neural Networks (CNNs) suffer a degradation of $-8.7\%$, caused by persistent destructive gradient conflict (cosine similarity $-0.26$). Crucially, strict capacity control experiments confirm

that these gains vanish when the auxiliary pathways are excluded from the loss, ruling out simple parameter expansion as the cause.

Our work makes three key contributions:

(i) We present the Asymmetric Training Paradigm as a controllable framework for analyzing architecture–objective interactions at both model and stage levels.

(ii) Using this probe, we identify Architectural Resonance—a stage-dependent phenomenon whereby auxiliary supervision efficacy varies with architectural inductive biases. We establish this exists on a resilience-modulated spectrum through within-model stage differentiation (CoAtNet on CIFAR-100) and model-level validation across architectures (ResNet/ViT on ImageNet-1k).

(iii) We provide quantitative characterization of the underlying mechanisms: (a) universal landscape smoothing at initialization, and (b) architecture-specific gradient dynamics during training, revealing how these jointly influence optimization and generalization.

## 2 RELATED WORK

### 2.1 AUXILIARY SUPERVISION AND MULTI-TASK LEARNING

The use of intermediate supervision is a well-established technique to combat vanishing gradients (Szegedy et al., 2015; Lee et al., 2015). Modern applications leverage auxiliary tasks for representation learning (Gidaris et al., 2018; Chen et al., 2020) and Multi-Task Learning (MTL) (Caruana, 1997; Ruder, 2017; Kendall et al., 2018). A critical challenge in these settings is gradient conflict, where competing objectives hinder optimization (Yu et al., 2020; Chen et al., 2018). Prior work has largely focused on mitigating such conflicts (e.g., Gradient Surgery (Yu et al., 2020)) or enforcing representational consistency within homogeneous objective families (Navon et al., 2022; Shamsian et al., 2023). In contrast, our study deliberately employs heterogeneous signals (non-competitive vs. competitive) not merely to enhance performance, but as a scientific probe to analyze how different architectures intrinsically respond to conflicting objectives. This perspective shifts the role of auxiliary supervision from a Performance-driven optimization aid to a lens for understanding architecture-specific optimization preferences.

### 2.2 ARCHITECTURAL INDUCTIVE BIASES

Our analysis is grounded in the distinct inductive biases of architectures (Goyal & Bengio, 2022). CNNs enforce strong priors on spatial locality and translation equivariance through weight-sharing kernels (LeCun et al., 1989; Cohen & Welling, 2016). In contrast, ViTs rely on self-attention for global relationships (Dosovitskiy et al., 2021; Vaswani et al., 2017), but this flexibility often comes at the cost of trainability, characterized by sharper optimization landscapes and higher sensitivity to initialization (Xiao et al., 2021; Chen et al., 2022). While these biases are well-studied in terms of optimization landscapes, feature geometries, and inductive bias mechanisms (Lu et al., 2022; Tuli et al., 2021), how they govern a model's response to heterogeneous supervisory signals remains largely unexplored. Our work addresses this by treating the supervisory signal as a controlled variable to probe these intrinsic architectural dynamics.

### 2.3 OPTIMIZATION LANDSCAPES AND REGULARIZATION

Understanding the geometry of the loss landscape is crucial for explaining generalization (Li et al., 2018; Keskar et al., 2017). Flatter minima are generally associated with better generalization, a principle leveraged by techniques like Sharpness-Aware Minimization (SAM) (Foret et al., 2021). Our findings contribute to this domain by identifying a structural mechanism for landscape smoothing. Following established methods for analyzing landscape geometry (Santurkar et al., 2019), we show that orthogonal auxiliary branches universally reduce initial gradient norms. However, this geometric benefit translates into performance gains only when auxiliary objectives resonate with architectural inductive biases. When objectives conflict with rigid spatial priors, gradient misalignment during training overrides the initial smoothing, leading to degradation despite improved landscape geometry.

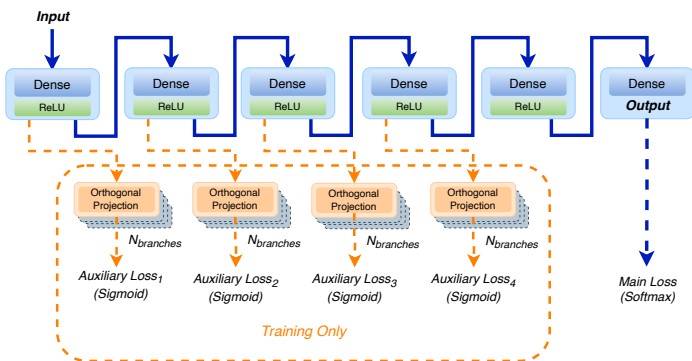

Figure 1: The Asymmetric Training Paradigm. Schematic overview using a simplified MLP architecture. Multiple auxiliary branches (orange) inject heterogeneous supervision (Sigmoid) into the backbone. These branches are removed at inference time, ensuring zero inference overhead.

# 3 METHODOLOGY

## 3.1 THE ASYMMETRIC TRAINING PARADIGM

We introduce the Asymmetric Training Paradigm (Figure 1), a controllable framework for probing architecture–objective interactions through structured gradient modulation. Unlike standard Multi-Task Learning which seeks to optimize multiple outputs, our paradigm uses auxiliary branches strictly as training-time scaffolding to analyze how different architectures respond to heterogeneous supervision. The framework is built on three pillars: **Asymmetry** (auxiliary branches are discarded at inference), **Heterogeneity** (auxiliary objectives differ qualitatively from the primary task), and **Controlled Redundancy** (scalable orthogonal pathways enabling systematic characterization of architectural resilience).

## 3.2 CORE HYPOTHESIS: ARCHITECTURAL RESONANCE

We propose the Architectural Resonance Hypothesis: The efficacy of heterogeneous auxiliary supervision depends on the compatibility between auxiliary signal characteristics and architectural inductive biases. This manifests as a spectrum of gradient interactions during training:

- **Constructive Interference:** When auxiliary signals align with an architecture's inductive bias (e.g., spatially-agnostic non-competitive signals for ViTs' global modeling capacity), they induce positive gradient alignment, enabling improved trainability and performance.

- **Destructive Interference:** When signals conflict with rigid structural priors (e.g., spatially-agnostic projections for CNNs' locality bias), they cause persistent gradient misalignment and performance degradation.

This reveals that auxiliary supervision efficacy is architecture-dependent: the degree of resonance determines whether heterogeneous signals enhance or impair training.

## 3.3 MECHANISM DESIGN AND CONTROLS

### 3.3.1 ARCHITECTURAL SETUP

To isolate the impact of inductive biases, we employ three controlled lightweight backbone architectures spanning a spectrum of inductive biases: a 6-layer MLP (minimal structural bias), a 6-conv-layer CNN with spatial downsampling (rigid locality bias), and a 6-block Vision Transformer (ViT-6L, flexible attention-based bias). To establish scalability, we extend our evaluation to ImageNet-1k using standard ResNet-18/50 and ViT-Small/B-16 backbones (Section 4.1). To validate stage-level mechanisms, we further analyze the hybrid CoAtNet architecture on CIFAR-100

(Section 4.2), which combines convolutional and transformer stages within a single model to enable within-network comparisons. Full architectural details are provided in Appendix C.3.

### 3.3.2 The Gradient Modulation Mechanism

A critical innovation in our method is separating parameter capacity from gradient flow. At each attachment point, we instantiate $N_{\text{branches}}$ parallel branches. We distinctly categorize them into two roles to isolate the source of gains: (1) **Static Gradient Modulators ($k = 1 \ldots N - 1$):** These branches are initialized orthogonally and then frozen (via $stop\_gradient$ applied to their projection weights). Crucially, while their weights $W_{aux}$ do not update ($\nabla_{W_{aux}} L = 0$), gradients still backpropagate through these fixed projections into the shared backbone features $h$ ($\nabla_h L \neq 0$). (2) **The Active Probe ($k = N$):** The final branch is fully trainable: unlike the frozen branches, its projection weights receive gradients and adapt during training. All auxiliary branches contribute Sigmoid/BCE losses, but only the last branch updates its own weights, while the earlier branches remain fixed orthogonal projections.

As a result, the backbone is optimized under a combination of rigid geometric constraints (from the static, orthogonal anchors whose gradients backpropagate through fixed projections) and adaptive error signals (from the active branch that continuously adjusts to the data). This design transforms these branches into fixed, structured "lenses" that refract the gradient flow without adding learnable capacity. By scaling $N$, we can systematically smooth the optimization landscape without the confounding factor of adding learnable parameters. As we empirically validate in Section 4.4 and Appendix A.1, this topology consistently reduces the initial gradient norm, creating a more favorable geometry for optimization.

### 3.3.3 Capacity Control Variant

To address the concern that improvements might stem simply from "adding more parameters" (even static ones), we define a Capacity Control baseline within our methodology. In this variant, we instantiate the exact same $N$ branches with identical initialization, but detach them from the loss graph. This ensures the model has the same parameter count and architecture, but no gradient modulation occurs. This control allows us to attribute performance gains strictly to the interaction of gradients, not the existence of parameters. Results are reported in Table 6.

### 3.3.4 Variables: Redundancy and Dialogue Strength

We manipulate two variables to map the resonance landscape:

- **Redundancy ($N_{\text{branches}}$):** The number of parallel branches per attachment point, denoted as $N\times$. This hyperparameter modulates the intensity of initial landscape smoothing. We evaluate performance regimes up to $20\times$ and extend to $300\times$ to probe architectural limits and analyze landscape mechanics.

- **Dialogue Strength ($\alpha$):** A scalar hyperparameter that balances the primary and auxiliary losses. Let $M$ be the number of attachment points (layers) and $N$ be the redundancy level (number of branches) at each point. The total training objective is:

$$L_{\text{total}} = L_{\text{main}} + \alpha \sum_{m=1}^{M} \sum_{n=1}^{N} L_{\text{aux}}^{(m,n)} \tag{1}$$

where $L_{\text{aux}}^{(m,n)}$ denotes the Sigmoid loss of the $n$-th branch at the $m$-th attachment point.

### 3.3.5 Heterogeneous Objectives

To induce the requisite "signal dialogue," we enforce heterogeneity between the objectives:

- **Primary (Competitive):** Standard Softmax Cross-Entropy ($L_{\text{main}}$), encouraging winner-takes-all feature discrimination.

- **Auxiliary (Coexistent):** Sigmoid Binary Cross-Entropy ($L_{\text{aux}}$), computed independently per class. This encourages the model to capture non-exclusive features for each class, fundamentally differing from the Softmax dynamic. We explicitly choose this heterogeneous

design because, as shown in our Heterogeneity Control experiments (Section 4.4), replacing Sigmoid with a homogeneous Softmax objective leads to performance degradation.

### 3.4 EVALUATION PROTOCOL: BALANCING RIGOR AND SCALABILITY

To ensure that our findings are both scientifically rigorous and practically applicable, we design a distinct evaluation protocol that adapts to the scale of the problem.

**Mechanistic Analysis (CIFAR)** For our controlled studies, our priority is isolating the source of gains. We employ a strict Two-Stage Tuning Protocol to rule out hyperparameter confounding: We first exhaustively tune the Baseline to find its optimal learning rate and weight decay. We then fix these backbone hyperparameters and search only for the dialogue strength $\alpha$ for our method.

**Large-Scale Validation (ImageNet)** For ImageNet-1k, we strictly maintain fixed backbone hyperparameters, attaching auxiliary heads only at predefined intermediate points (Table 21). Optimization follows standard protocols: linear scaling SGD for ResNet (Goyal et al., 2018) and AdamW for ViT (adapted from (Touvron et al., 2021) using a clean baseline protocol). By tuning only $\alpha$ (on a held-out subset), we assess the method's scalability in realistic, compute-constrained scenarios.

Crucially, across all experimental settings, including the targeted hybrid verification on CoAtNet (Dai et al., 2021), backbone hyperparameters are kept strictly identical between the baseline and our method. This guarantees that any observed performance difference, whether improvement or degradation, is attributable solely to the proposed topological interaction, ruling out hyperparameter mismatch as a confounder.

## 4 EXPERIMENTS

### 4.1 MAIN RESULTS: THE DIVERGENCE OF VIT AND CNN

We first evaluate the Asymmetric Training Paradigm on CIFAR-100 using three representative architectures: MLP, CNN, and ViT-6L across 10 random seeds. As shown in Table 1, the impact of our heterogeneous auxiliary supervision is fundamentally architecture-dependent.

**Dose-Response on CIFAR-100** We observe a clear correlation between the level of redundancy ($N_{\text{branches}}$) and performance modulation (Figure 2), revealing distinct architectural preferences:

- **Vision Transformers (Constructive Synergy):** ViT-6L benefits significantly, with accuracy improving monotonically as redundancy increases. At $20\times$ redundancy, it achieves a +9.2% improvement in top-1 accuracy over the baseline. This provides evidence that ViTs can constructively integrate dense heterogeneous signals.

- **CNNs (Destructive Conflict):** Conversely, the CNN suffers severe degradation across all redundancy levels, experiencing up to a -15.4% decrease in accuracy. This indicates a fundamental incompatibility between spatially-agnostic Sigmoid signals and the CNN's strong locality priors.

- **MLPs (Inconsistent/Noisy Interaction):** The MLP exhibits a distinct behavior. While a single auxiliary branch ($1\times$) provides mild regularization (+1.1%), increasing redundancy leads to consistent degradation (e.g., -2.1% at $20\times$). This suggests that without structural mechanisms such as self-attention to align auxiliary signals, MLPs are destabilized by gradient noise as redundancy scales.

**Scalability on ImageNet-1k** To confirm that the observed divergence is not limited to small-scale datasets, we extend our evaluation to ImageNet-1k using ResNet-18/50 and ViT-Small/B-16 across 4 random seeds (Table 2). We strictly adhere to established training recipes (SGD for ResNet, AdamW for ViT) while adopting a "clean baseline" setup—standard augmentation without heavy regularization (e.g., Mixup, CutMix). The only structural difference between the Baseline and Asymmetric models is the addition of orthogonally initialized auxiliary heads. Both share identical backbones, data pipelines, and optimization hyperparameters, details are provided in Appendix C.4. Crucially,

Table 1: Architecture performance comparison across different configurations on CIFAR-100

| $N_{branches}$ | Arch | Plain | Asymmetric | p-value |
|---|---|---|---|---|
| 1× | MLP | 23.2 ± 0.3 | 24.3 ± 0.3 (+1.1; $\alpha = 100.0$) | < 0.001 |
| | CNN | 39.5 ± 0.4 | 30.8 ± 0.3 (-8.7; $\alpha = 100.0$) | < 0.001 |
| | ViT-6L | 35.8 ± 1.0 | 35.9 ± 0.5 (+0.1; $\alpha = 4.642$) | 0.7773 |
| 7× | MLP | 23.4 ± 0.5 | 22.1 ± 0.3 (-1.3; $\alpha = 0.1$) | < 0.001 |
| | CNN | 39.8 ± 0.6 | 31.7 ± 2.9 (-8.1; $\alpha = 0.1$) | < 0.001 |
| | ViT-6L | 35.9 ± 0.8 | 40.0 ± 2.1 (+4.1; $\alpha = 4.642$) | < 0.001 |
| 10× | MLP | 23.3 ± 0.3 | 21.7 ± 0.2 (-1.6; $\alpha = 0.1$) | < 0.001 |
| | CNN | 39.7 ± 0.6 | 24.3 ± 2.1 (-15.4; $\alpha = 1.0$) | < 0.001 |
| | ViT-6L | 36.0 ± 0.9 | 41.8 ± 2.2 (+5.8; $\alpha = 4.642$) | < 0.001 |
| 20× | MLP | 23.2 ± 0.3 | 21.1 ± 0.3 (-2.1; $\alpha = 0.1$) | < 0.001 |
| | CNN | 39.7 ± 0.6 | 33.6 ± 1.2 (-6.1; $\alpha = 0.1$) | < 0.001 |
| | ViT-6L | 36.1 ± 1.0 | 45.3 ± 1.3 (+9.2; $\alpha = 1.0$) | < 0.001 |

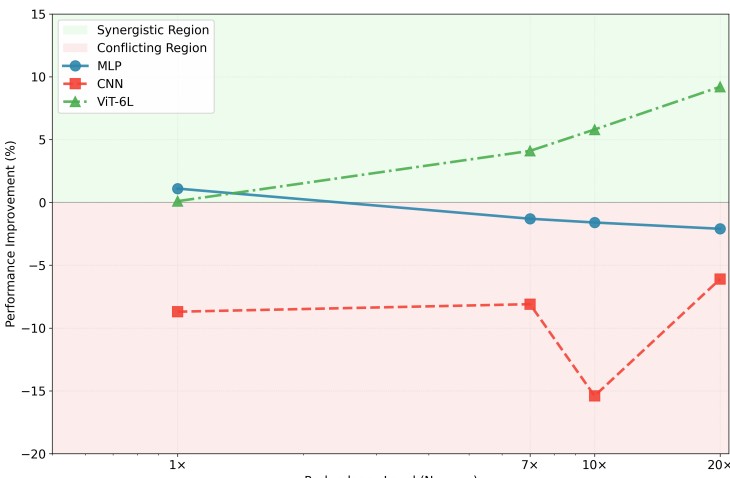

Figure 2: Architecture-dependent dose-response to auxiliary supervision. Each architecture shows distinct sensitivity patterns: ViT (constructive synergy), CNN (destructive conflict), MLP (inconsistent/noisy interaction). Performance improvement plotted against branch redundancy ($N_{\text{branches}}$) on CIFAR-100.

to rule out hyperparameter hacking, we tuned the dialogue strength $\alpha$ on a 10% held-out training subset, ensuring the validation set remained strictly unseen during the search. This rigorous setup guarantees that observed gains stem purely from structural resonance rather than overfitting or data-level artifacts. Consistent with CIFAR-100, ViT backbones show robust improvements (e.g., +2.25% on ViT-B/16 at 20× redundancy, $p < 0.001$), suggesting that constructive resonance scales to large-scale benchmarks. In contrast, ResNets remain neutral. This suggests that architectural resilience mechanisms (e.g., residual connections) may mitigate the gradient conflicts that caused plain CNNs to collapse, thereby reinforcing the architectural dependence of the phenomenon.

While ViTs consistently benefit from auxiliary resonance, CNNs exhibit either destructive conflict (Plain CNN) or neutrality (ResNet). Given these distinct architectural preferences, a critical question arises: how do hybrid architectures behave? We address this in Section 4.2.

## 4.2 TARGETED VERIFICATION ON HYBRID ARCHITECTURE (COATNET)

To empirically validate the stage-dependent Efficacy of our Architectural Resonance hypothesis, we extend our analysis to CoAtNet (Dai et al., 2021), a hybrid architecture that integrates convolutional

Table 2: Architecture performance comparison on ImageNet-1k

| Architecture | Baseline | 1× | 10× | 20× |
|---|---|---|---|---|
| ResNet-18 | 68.23±0.20 | 68.33±0.10 | 68.29±0.10 | 68.37±0.13 |
| ResNet-50 | 73.83±0.09 | 73.55±0.17** | 73.75±0.11 | 73.76±0.21 |
| ViT-Small | 69.13±0.14 | 70.16±0.09*** | 70.27±0.18** | 70.35±0.29** |
| ViT-B/16 | 66.75±0.13 | 68.34±0.64* | 68.83±0.27*** | 69.00±0.19*** |

*Statistical significance:* $* p < 0.05$, $** p < 0.01$, $*** p < 0.001$.

stages (early) and transformer stages (late). This allows us to test whether the observed divergence is strictly stage-dependent.

We employ a CIFAR-100 adapted CoAtNet-0 backbone to strictly preserve the original multi-stage layout while adapting to the dataset resolution. We apply our asymmetric probe separately to the Convolutional Stage (S2) and the Transformer Stage (S3) across varying redundancy levels ($N_{branches}$) to isolate their differential responses.

As presented in Table 3, the results reveal an observable divergence between stages within the same model: (1) **Synergy in Transformer Stages (S3):** Consistent with our main ViT results, applying auxiliary supervision to S3 yields consistent improvements. It achieves a peak accuracy of 77.08% (+1.74%), showing statistically significant resonance ($p < 0.001$). (2) **Conflict in Convolutional Stages (S2):** In contrast, applying dense probes to S2 shows limited compatibility. While mild redundancy (20×) exhibits marginal response, high redundancy (300×) leads to a statistically significant tendency toward conflict (-0.80%, $p < 0.05$).

These findings provide targeted empirical support for our Architectural Resonance Hypothesis. Comparing CoAtNet S2 with the plain CNN (Table 1) reveals the role of architectural resilience: modern components such as residual blocks and batch normalization help buffer against immediate collapse (S2: neutral at 1× vs. CNN: -8.7%), yet degradation still emerges under extreme redundancy (S2: -0.8% at 300×). This confirms that auxiliary supervision efficacy is not binary but stage-dependent: attention-based stages benefit consistently from heterogeneity, whereas convolutional stages exhibit limited compatibility—delayed by modern architectural features but fundamentally prone to conflict.

Table 3: Stage-Dependent Response on CIFAR-100 adapted CoAtNet-0. Comparison of applying auxiliary supervision to CNN (S2) vs. Transformer (S3) stages. Results averaged over 5 random seeds.

| Stage | 1× | 20× | 100× | 300× |
|---|---|---|---|---|
| S2 (CNN) | 75.26±0.59 (-0.08) | 75.88±0.14 (+0.54**) | 75.59±0.44 (+0.25) | **74.54±0.48** (**-0.80**\*) |
| S3 (Transformer) | 75.91±0.28 (+0.57**) | 76.76±0.37 (+1.42***) | **77.08±0.27** (**+1.74**\***) | 76.82±0.30 (+1.48***) |
| Baseline | $75.34 \pm 0.20$ | | | |
| Phenomenon | ViT starts gaining | Strong Synergy (S3) | **Peak (S3)** | **Conflict (S2)** |

*Statistical significance:* $* p < 0.05$, $** p < 0.01$, $*** p < 0.001$.

### 4.3 UNDERLYING DYNAMICS: LANDSCAPE GEOMETRY AND GRADIENT FLOW

Having confirmed the performance gains, we now investigate the mechanistic cause. We analyze the training dynamics from two perspectives: the initial geometry of the optimization landscape and the directional alignment of gradients during training.

**Initial Phase: Landscape Smoothing**  As hypothesized in Section 3, the static orthogonal branches are designed to condition the loss landscape. Following (Santurkar et al., 2019), we quantify this by measuring the Initial Gradient Norm of the main loss across 5 random seeds.

- **Observation:** Table 4 presents the results. Increasing redundancy ($N_{\text{branches}}$) drastically reduces the gradient norm across all architectures. Notably, for MLP and CNN, $300\times$ redundancy reduces the gradient norm by over 90%.

- **Implication:** This confirms that our static topology acts as a "geometric conditioner," creating a smoother initial surface. This is particularly beneficial for ViTs, which are known to suffer from sharp, ill-conditioned landscapes (Chen et al., 2022).

Table 4: Landscape Smoothing Effect

| Architecture | $1\times$ | | $20\times$ | | $300\times$ | |
|---|---|---|---|---|---|---|
| | Result | p-value | Result | p-value | Result | p-value |
| MLP | -29.86±20.35 | 0.0305 | -65.38±1.95 | < 0.001 | -90.40±0.72 | < 0.001 |
| ViT-6L | -3.40±0.47 | < 0.001 | -33.68±3.62 | < 0.001 | -68.78±8.85 | < 0.001 |
| CNN | -16.64±4.79 | 0.0015 | -67.22±5.52 | < 0.001 | -90.18±2.38 | < 0.001 |

**Training Phase: Gradient Alignment**  While landscape smoothing is generally beneficial, why do CNNs degrade under the same condition? To answer this, we analyze the Cosine Similarity between the gradients of the main task ($L_{\text{main}}$) and the auxiliary task ($L_{\text{aux}}$) throughout training (Figure 3 and Table 5).

- **Constructive Synergy (ViT):** ViTs exhibit positive cosine similarity (mean $+0.19$), indicating that the auxiliary signals point in a direction compatible with the main objective. This alignment allows the ViT to effectively exploit the smoothed landscape established in the initial phase, translating geometric potential into performance gains.

- **Destructive Conflict (CNN):** Despite the smoothed landscape, CNNs show persistent negative similarity, reaching as low as $-0.82$ (mean $-0.26$). This suggests that Sigmoid signals fundamentally conflict with the CNN's locality-biased kernels. This intense gradient conflict overrides the benefits of smoothing, turning the auxiliary signals into destructive interference.

- **Inconsistent / Noisy Interaction (MLP):** The MLP trajectory fluctuates without a consistent direction, alternating between weak positive and negative values (e.g., oscillating between $+0.07$ and $-0.34$). Lacking strong inductive biases to orient these auxiliary signals, the interaction is effectively incoherent. Consequently, as redundancy increases, these unaligned signals accumulate as gradient noise rather than constructive guidance, explaining the degradation observed in Table 1.

Table 5: Detailed Gradient Conflict Analysis

| Architecture | Final Sim | Avg Sim | Min Sim | Max Sim | Gradient Interaction |
|---|---|---|---|---|---|
| MLP | -0.0309 | -0.0801 | -0.3422 | 0.0701 | Inconsistent / Noisy Interaction |
| CNN | -0.1926 | -0.2574 | -0.8210 | 0.0464 | Destructive Conflict |
| ViT | 0.2631 | 0.1870 | -0.1845 | 0.3654 | Constructive synergy |

**Visualizing the Resonance: Attention Maps**  To visibly corroborate the gradient alignment findings, we visualize the self-attention maps of the final CLS token (Figure 4). Compared to the Baseline, the Asymmetric ViT exhibits significantly sharper attention on the semantic object foreground, filtering out background noise. This confirms that the constructive gradient synergy translates into more semantic feature extraction.

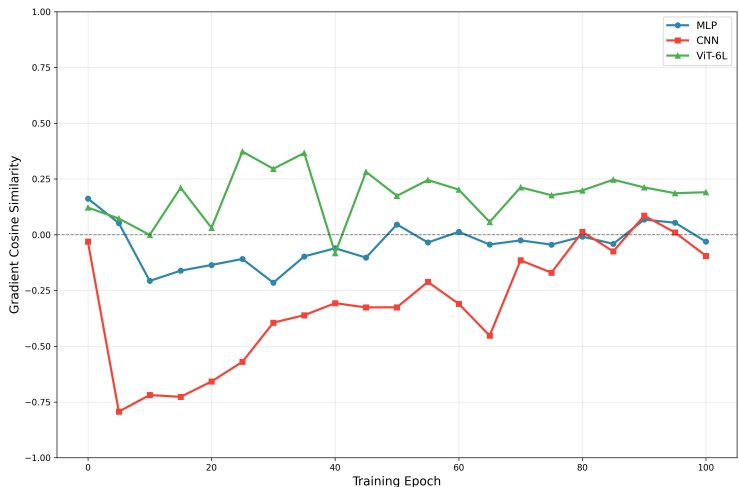

Figure 3: Gradient conflict evolution across architectures during training

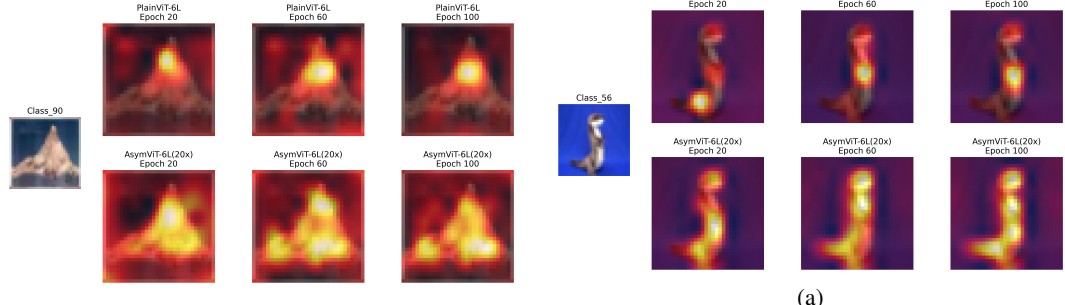

(a)

Figure 4: Visualizing Architectural Resonance. Attention maps show that Asymmetric Training induces sharper focus on the object compared to Baseline (PlainViT-6L), confirming constructive feature learning.

### 4.4 MECHANISM VERIFICATION: RULING OUT CONFOUNDERS

A critical question is identifying the source of these gains. Is the improvement in ViTs driven by the proposed topological interaction, or simply by parameter expansion (capacity) or generic deep supervision? We address this through strictly controlled ablation studies (10 random seeds).

**Capacity Control** Is it just adding parameters? We compare against a Capacity Control baseline where the same $N$ auxiliary branches are instantiated but detached from the loss graph (weights present but gradients blocked). As shown in Table 6 (Top), mere parameter redundancy yields zero statistical gain ($p > 0.05$). For ViT-6L, increasing branches to $20\times$ in the control group results in negligible fluctuation, whereas our active method achieves $+9.2\%$ improvement. This rigorously rules out implicit regularization from parameter count as the cause. The gain stems from the interaction of gradients.

**Heterogeneity Control** Is it just Deep Supervision? We investigate the necessity of Heterogeneity by replacing our Sigmoid auxiliary loss with a standard Softmax auxiliary loss (Homogeneous). Table 6 (Middle) reveals that homogeneous supervision degrades performance for ViTs (e.g., $-5.3\%$ drop). This confirms that ViTs specifically benefit from the "non-competitive" nature of the Sigmoid signal to facilitate capacity exploitation.

**Random Init Control** Is it just initialization noise? We compare our Orthogonal Initialization strategy against standard Random Initialization for the auxiliary branches. As shown in Table 6

(Bottom), using Random Initialization fails to provide consistent gains and often leads to training instability (high variance). This suggests that the structural orthogonality is a prerequisite for effective resonance. The auxiliary branches must be geometrically aligned (via orthogonality) to probe the landscape constructively, rather than injecting unstructured noise.

Table 6: Ablation study results across different control conditions on CIFAR-100. This table investigates potential confounders. **Top:** Adding parameters without gradient flow yields no gain. **Middle:** Replacing Sigmoid with Softmax (Homogeneous) causes degradation. **Bottom:** Using Random Initialization instead of Orthogonal leads to instability (e.g., ViT -4.0%). **Contrast:** Our Asymmetric method (Sigmoid + Orthogonal) achieves **+9.2%** absolute improvement on ViT under the same $20\times$ condition (Table 1).

| Control Type | $N_{branches}$ | MLP | CNN | ViT-6L |
|---|---|---|---|---|
| Capacity Control | $1\times$ | 23.2±0.3 (-0.1) | 39.9±0.7 (+0.7) | 36.1±1.1 (+0.1) |
| | $10\times$ | 23.1±0.4 (-0.0) | 39.8±0.8 (-0.1) | 35.8±1.2 (-0.3) |
| | $20\times$ | 23.1±0.4 (-0.0) | 40.0±1.0 (+0.4) | 36.1±1.1 (-0.0) |
| Heterogeneity Control | $1\times$ | 24.1±0.3 (+0.8**) | 37.4±0.4 (-2.1***) | 36.1±0.5 (+0.2) |
| | $10\times$ | 20.4±0.6 (-2.7***) | 36.2±0.9 (-3.4***) | 33.9±0.6 (-1.9***) |
| | $20\times$ | 19.0±0.5 (-4.3***) | 37.9±1.0 (-1.9***) | 35.7±0.9 (-0.3) |
| Random Init. Control | $1\times$ | 24.3±0.4 (+1.0***) | 31.6±0.7 (-7.9***) | 36.3±0.9 (+0.3) |
| | $10\times$ | 22.8±0.3 (-0.3) | 30.6±1.0 (-9.6***) | 35.6±1.1 (-0.5) |
| | $20\times$ | 23.0±0.2 (-0.3) | 30.8±0.8 (-9.1***) | 31.8±1.1 (-4.0***) |

*Statistical significance: * $p < 0.05$, ** $p < 0.01$, *** $p < 0.001$.*

## 5 DISCUSSION AND LIMITATIONS

Our Asymmetric Training Paradigm serves as a probe for architecture–objective interactions, yielding measurable gains in the tested ViT backbones and offering mechanistic insights into gradient dynamics.

**Implications** Crucially, targeted verification on CoAtNet (Section 4.2) advances our understanding beyond binary outcomes. It suggests a Stage-Adaptive Strategy for modern model design: heterogeneous auxiliary signals may be most effective in attention-based stages, while requiring caution in convolutional stages to avoid structural conflict. This challenges the assumption that "more supervision is always better."

**Limitations** Despite these findings, we acknowledge several limitations: (1) Convolutional Compatibility: CNN stages in our experiments show limited compatibility with dense heterogeneous probes, reflecting the rigidity of spatial inductive biases. This currently restricts applicability to attention-based backbones; (2) Training Overhead: While incurring zero inference overhead, training costs scale linearly with redundancy ($N_{branches}$), creating memory pressure for extremely large-scale pre-training; (3) Hyperparameter Search: The current approach relies on a grid search to identify the optimal dialogue strength $\alpha$, which is time-consuming and may yield sub-optimal configurations compared to dynamic schedules; (4) Theoretical Formalism: While we provide empirical evidence and mechanistic analysis, a closed-form theoretical framework quantifying the relationship between orthogonality, redundancy, and generalization gap remains an open challenge; (5) Scope: Whether Architectural Resonance extends to other modalities (e.g., NLP) or loss types requires verification.

**Future Work** We envision five key directions: (1) designing spatially-aware signals that respect CNN locality; (2) developing adaptive modulation strategies for $\alpha$ to eliminate expensive search; (3) distilling the geometric smoothing effect of redundancy into computationally efficient, analytically equivalent formulations to reduce memory pressure; (4) mathematically formalizing resonance conditions; and (5) extending the paradigm to diverse domains to test universality beyond vision.

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

# A APPENDIX

## A.1 LOSS LANDSCAPE ANALYSIS

We conducted systematic loss landscape analysis across MLP, CNN, and ViT-6L architectures on CIFAR-10 and CIFAR-100. Following established protocols (Li et al., 2018), we visualized the loss surfaces using a 51×51 grid centered at the initialization point, with directions determined by random Gaussian perturbations normalized to unit variance. To balance computational efficiency with statistical reliability, we randomly sampled 500 training instances for loss evaluation at each grid point. This sampling size provides sufficient statistical power while remaining computationally tractable for systematic analysis across multiple architectures and redundancy levels. The resulting visualizations reveal distinct architectural signatures in terms of loss surface smoothness and optimization landscape complexity. (Figure 5, Table 7 and 8)

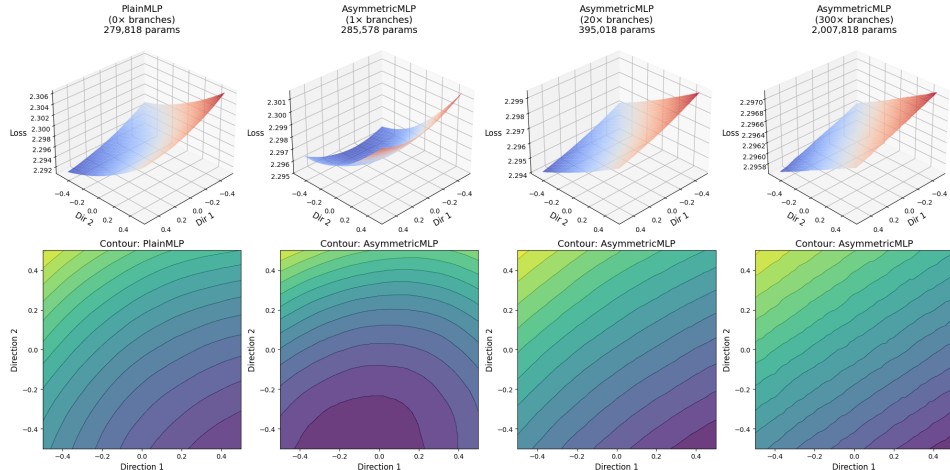

Figure 5: Loss landscape visualization for MLP on CIFAR-10

Table 7: Progressive loss landscape smoothing in MLP architecture on CIFAR-10. Standard deviation (Std), range, and mean gradient magnitude all decrease systematically with increased redundancy, demonstrating that topological modifications consistently flatten the optimization surface independent of final performance outcomes. Percentages indicate relative change from baseline.

| Model | Params | Std(Loss) | Range(Loss) | Mean(Grad) |
|---|---|---|---|---|
| Plain | 0.28M | 0.0032 | 0.0152 | 0.0002 |
| Asymmetric(1×) | 0.29M | 0.0015 (-52.3%) | 0.0064 (-57.6%) | 0.0001 (-49.8%) |
| Asymmetric(20×) | 0.40M | 0.0011 (-64.4%) | 0.0053 (-64.9%) | 0.0001 (-65.0%) |
| Asymmetric(300×) | 2.01M | 0.0003 (-90.2%) | 0.0015 (-90.3%) | 0.0000 (-90.4%) |

# B BOUNDARY CONDITIONS AND EXTENDED ANALYSIS

## B.1 SAMPLE EFFICIENCY AND GENERALIZATION IN LOW-DATA REGIMES

To assess the generalizability of our architectural resonance findings under data-scarce conditions, we conducted systematic few-shot learning experiments on CIFAR-10 and CIFAR-100. We hypothesize that asymmetric training benefits should be amplified in low-data regimes, where auxiliary supervision can provide crucial structural guidance when primary signals are sparse (Tables 9, 10, and 11).

**Experimental Design.** We systematically varied the number of training samples per class from 5 to 5000, creating a comprehensive data scarcity spectrum. For each data regime, we maintained the original test set size to ensure consistent evaluation conditions. All experiments were repeated

Table 8: Cross-architecture comparison of loss landscape smoothing on CIFAR-100. Despite universal landscape flattening effects (up to 90% reduction in surface roughness), architectural differences emerge: CNN shows the most dramatic smoothing with minimal parameter increase, while ViT-6L exhibits more modest but consistent improvements. These results demonstrate that landscape conditioning is architecture-agnostic, yet performance benefits depend critically on architectural resonance with auxiliary signals.

| Model | Params | Std(Loss) | Range(Loss) | Mean(Grad) |
|---|---|---|---|---|
| **CNN** | | | | |
| Plain | 2.43M | 0.0008 | 0.0040 | 0.0001 |
| Asymmetric(1×) | 2.51M | 0.0007 (-16.3%) | 0.0033 (-17.3%) | 0.0000 (-15.9%) |
| Asymmetric(20×) | 4.09M | 0.0003 (-66.7%) | 0.0013 (-68.0%) | 0.0000 (-66.9%) |
| Asymmetric(300×) | 27.4M | 0.0001 (-90.2%) | 0.0004 (-90.3%) | 0.0000 (-90.2%) |
| **ViT-6L** | | | | |
| Plain | 1.22M | 0.0045 | 0.0212 | 0.0003 |
| Asymmetric(1×) | 1.29M | 0.0043 (-3.7%) | 0.0204 (-3.7%) | 0.0003 (-3.6%) |
| Asymmetric(20×) | 2.75M | 0.0028 (-37.7%) | 0.0132 (-38.0%) | 0.0002 (-37.1%) |
| Asymmetric(300×) | 24.3M | 0.0009 (-80.1%) | 0.0044 (-79.3%) | 0.0001 (-77.6%) |

across 10 random seeds with stratified sampling to ensure class balance. Statistical significance was assessed using two-tailed paired t-tests.

**Theoretical Motivation.** Under data scarcity, the auxiliary sigmoid branches should provide particularly valuable regularization, as the primary softmax objective becomes increasingly prone to overfitting. This effect should be most pronounced in architectures that exhibit gradient synergy rather than conflict.

Table 9: Few-shot learning performance of MLP on CIFAR-10.

| Samples/Class | PlainMLP | AsymmetricMLP | Improvement | p-value |
|---|---|---|---|---|
| 10 | 22.90 ± 1.67 | 22.99 ± 1.71 | +0.09 | 0.8082 |
| 50 | 28.70 ± 1.00 | 28.97 ± 0.82 | +0.27 | 0.4346 |
| 100 | 31.99 ± 1.18 | 33.30 ± 0.93 | +1.31 | 0.0122 |
| 500 | 38.93 ± 1.17 | 39.44 ± 1.05 | +0.51 | 0.3087 |
| 1000 | 43.13 ± 0.39 | 44.00 ± 0.43 | +0.87 | 0.0043 |
| 5000 | 53.57 ± 0.39 | 53.74 ± 0.56 | +0.17 | 0.4824 |

Table 10: Few-shot learning performance of MLP on CIFAR-100.

| Samples/Class | PlainMLP | AsymmetricMLP | Improvement | p-value |
|---|---|---|---|---|
| 5 | 4.06 ± 0.64 | 5.22 ± 0.25 | +1.16 | 0.0002 |
| 10 | 7.27 ± 0.38 | 6.97 ± 0.39 | -0.30 | 0.0619 |
| 20 | 9.36 ± 0.36 | 9.49 ± 0.29 | +0.13 | 0.2965 |
| 50 | 12.98 ± 0.45 | 13.98 ± 0.48 | +1.00 | 0.0041 |
| 100 | 15.95 ± 0.28 | 16.97 ± 0.29 | +1.02 | 0.0000 |
| 200 | 21.31 ± 0.44 | 22.25 ± 0.22 | +0.94 | 0.0004 |
| 500 | 25.94 ± 0.31 | 26.65 ± 0.32 | +0.71 | 0.0007 |

## B.2 GENERALIZATION STABILITY UNDER LABEL CORRUPTION

We evaluated model resilience under label noise by corrupting a fraction of training labels and measuring performance degradation. Label noise was introduced by randomly flipping labels with probabilities ranging from 10% to 90%, while maintaining the original test set for consistent evaluation (Tables 13 and 14).

Table 11: Few-shot learning performance of ViT-6L(20×) on CIFAR-100.

| Samples/Class | PlainViT-6L | AsymmetricViT-6L | Improvement | p-value |
|---|---|---|---|---|
| 5 | 5.10 ± 0.50 | 5.59 ± 0.33 | +0.49 | 0.0022 |
| 10 | 7.92 ± 0.52 | 8.79 ± 0.49 | +0.87 | 0.0008 |
| 20 | 9.67 ± 0.49 | 10.66 ± 0.68 | +0.99 | 0.0052 |
| 50 | 15.61 ± 0.79 | 15.58 ± 0.51 | -0.03 | 0.8687 |
| 100 | 20.29 ± 0.58 | 23.16 ± 0.78 | +2.87 | 0.0001 |
| 200 | 28.00 ± 0.62 | 32.23 ± 1.79 | +4.23 | 0.0011 |
| 500 | 43.53 ± 1.45 | 53.28 ± 0.65 | +9.75 | 0.0000 |

Table 12: Architecture Performance Comparison (CIFAR-10)

| Architecture | Baseline | Asymmetric(1×) | Improvement | P-Value |
|---|---|---|---|---|
| MLP | 49.5 ± 0.3 | 50.6 ± 0.4 | +1.1 | 0.0001 |
| CNN | 77.0 ± 0.6 | 77.2 ± 1.0 | +0.2 | 0.5371 |
| ViT-6L | 62.5 ± 0.7 | 63.6 ± 0.4 | +1.1 | 0.0068 |

# C  DETAILED EXPERIMENTAL CONFIGURATION

## C.1  HYPERPARAMETER SETTINGS

All hyperparameters were determined through systematic grid search following our "Pragmatic Gold Standard" strategy to ensure fair comparison. This three-stage optimization process isolates the effect of our asymmetric training paradigm while maintaining scientific rigor.

### C.1.1  OPTIMIZATION STRATEGY

For each architecture, we employed a principled three-stage hyperparameter search:

**Stage 1: Learning Rate Optimization** We fixed weight decay at $10^{-4}$ and conducted grid search over learning rates $\{10^{-4}, 3 \times 10^{-4}, 10^{-3}, 3 \times 10^{-3}, 10^{-2}\}$ for the baseline Plain model, training for 150 epochs and selecting the configuration yielding highest validation accuracy.

**Stage 2: Weight Decay Refinement** Using the optimal learning rate from Stage 1, we searched over weight decay values $\{10^{-1}, 10^{-2}, 10^{-3}, 10^{-4}, 10^{-5}\}$ for the Plain model, again training for 150 epochs.

**Stage 3: Auxiliary Weight Search** With optimal learning rate and weight decay fixed, we searched for the optimal auxiliary weight $\alpha$ using logarithmic spacing: $\alpha \in \{0.1, 0.215, 0.464, 1.0, 2.154, 4.642, 10.0, 21.544, 46.416, 100.0\}$ for the Asymmetric model. For CIFAR-10 MLP specifically, we employed linear spacing $\alpha \in [0, 50]$ to accommodate its distinct optimization characteristics.

### C.1.2  FINAL HYPERPARAMETER CONFIGURATIONS

The optimal hyperparameters (CIFAR-100) determined through our systematic search are showed in Table 15.

## C.2  TRAINING CONFIGURATION

**Training Duration:** All final results were obtained using 200 epochs.

**Statistical Validation:** Each configuration was evaluated across 10 independent runs with different random seeds (42-51) to ensure statistical reliability. Performance comparisons used two-tailed paired t-tests.

Table 13: MLP performance under label noise on CIFAR-10

| Noise Level | PlainMLP | AsymmetricMLP | Improvement | P-Value |
|---|---|---|---|---|
| 0.0% | 53.52 ± 0.53 | 53.74 ± 0.36 | +0.22 | 0.2025 |
| 10.0% | 51.82 ± 0.47 | 52.08 ± 0.42 | +0.26 | 0.1785 |
| 20.0% | 49.92 ± 0.55 | 50.22 ± 0.70 | +0.30 | 0.3667 |
| 30.0% | 47.69 ± 0.36 | 48.42 ± 0.45 | +0.73 | 0.0007 |
| 40.0% | 44.96 ± 0.62 | 45.90 ± 0.85 | +0.94 | 0.0067 |
| 50.0% | 42.13 ± 1.09 | 42.66 ± 0.52 | +0.53 | 0.2633 |
| 60.0% | 38.35 ± 0.57 | 39.23 ± 0.70 | +0.88 | 0.0530 |
| 70.0% | 32.94 ± 0.85 | 32.05 ± 0.93 | -0.89 | 0.0291 |
| 80.0% | 23.69 ± 1.61 | 22.94 ± 0.94 | -0.75 | 0.1580 |
| 90.0% | 10.43 ± 0.53 | 10.13 ± 0.61 | -0.30 | 0.1081 |

Table 14: MLP performance under label noise on CIFAR-100

| Noise Level | PlainMLP | AsymmetricMLP | Improvement | P-Value |
|---|---|---|---|---|
| 0.0% | 25.64 ± 0.29 | 26.00 ± 0.34 | +0.36 | 0.0040 |
| 10.0% | 24.76 ± 0.36 | 25.36 ± 0.47 | +0.60 | 0.0104 |
| 20.0% | 23.83 ± 0.34 | 24.29 ± 0.38 | +0.46 | 0.0014 |
| 30.0% | 22.54 ± 0.25 | 23.09 ± 0.36 | +0.55 | 0.0006 |
| 40.0% | 20.97 ± 0.42 | 21.82 ± 0.36 | +0.85 | 0.0023 |
| 50.0% | 19.08 ± 0.43 | 20.15 ± 0.28 | +1.07 | 0.0000 |
| 60.0% | 16.52 ± 0.39 | 17.97 ± 0.44 | +1.45 | 0.0000 |
| 70.0% | 12.76 ± 0.63 | 14.96 ± 0.58 | +2.20 | 0.0000 |
| 80.0% | 8.16 ± 0.73 | 9.56 ± 0.49 | +1.40 | 0.0005 |
| 90.0% | 3.60 ± 0.54 | 3.49 ± 0.43 | -0.11 | 0.4902 |

**Hardware:** All experiments were conducted on NVIDIA RTX 3090 GPUs with consistent computational environments to ensure reproducibility.

## C.3 ARCHITECTURE-SPECIFIC DETAILS

**MLP:** 6 linear layers with ReLU activations. Auxiliary branches attached after the first 4 ReLU activations.

**CNN:** 6 convolutional layers, 2 MaxPooling layers, 1 Dropout layer, and 3 linear layers. Auxiliary branches are strategically placed after ReLU activations in convolutional blocks. When a convolutional layer is immediately followed by max-pooling, the auxiliary branch is placed after the max-pooling operation to maintain spatial coherence.

**ViT-6L:** 6 Transformer blocks with 4 attention heads each and embedding dimension of 128. Auxiliary branches attached after each Transformer block output.

All auxiliary branches consist of a single linear layer with output dimension equal to the number of classes, initialized using orthogonal initialization for training stability.

## C.4 IMAGENET EXPERIMENTS

In this section, we provide documentation for ImageNet-1k. (From Table 16 to Table 23)

## C.5 COATNET EXPERIMENTS

This provides comprehensive details of the experimental setup used in our baseline experiments. All configurations follow standard practices in modern CIFAR-100 image classification research.

Table 15: Optimal hyperparameters for Table 1

| $N_{branches}$ | Architecture | Learning Rate | Weight Decay | $\alpha$ |
|---|---|---|---|---|
| | MLP | 0.0001 | 0.001 | 100.0 |
| $1\times$ | CNN | 0.0003 | 0.01 | 100.0 |
| | ViT-6L | 0.001 | 0.1 | 4.642 |
| | MLP | 0.0001 | 0.001 | 0.1 |
| $7\times$ | CNN | 0.0003 | 0.001 | 0.1 |
| | ViT-6L | 0.001 | 0.01 | 4.642 |
| | MLP | 0.0001 | 0.001 | 0.1 |
| $10\times$ | CNN | 0.0003 | 0.001 | 1.0 |
| | ViT-6L | 0.001 | 0.01 | 4.642 |
| | MLP | 0.0001 | 0.001 | 0.1 |
| $20\times$ | CNN | 0.0003 | 0.001 | 0.1 |
| | ViT-6L | 0.001 | 0.01 | 1.0 |

Table 16: Optimal alpha values for Random Seed 42

| Architecture | 1X | 10X | 20X |
|---|---|---|---|
| ResNet-18 | 0.0178 | 0.2371 | 0.0237 |
| ViT-Small | 23.7137 | 0.0237 | 0.0237 |
| ResNet-50 | 0.0750 | 0.0042 | 0.0178 |
| ViT-B/16 | 23.7137 | 0.0237 | 0.0042 |

### C.5.1 HYPERPARAMETER CONFIGURATION

Table 24 summarizes the complete set of hyperparameters used in our baseline experiments.

### C.5.2 MODEL ARCHITECTURE DETAILS

Our baseline model uses a CIFAR-100 adapted CoAtNet backbone, structurally similar to CoAtNet-0 but scaled down for the $32 \times 32$ input resolution. Table 25 details the stage-wise configuration.

The architecture follows the Conv–Conv–Attention–Attention pattern proposed in the original CoAt-Net paper (Dai et al., 2021), where early stages use convolutional MBConv blocks and later stages employ Transformer blocks. The transition from convolution to attention occurs between S2 and S3.

### C.5.3 JUSTIFICATION OF CONFIGURATION CHOICES

**Training budget:** The 200-epoch training with batch size 128 aligns with common practice in ResNet and ViT works on CIFAR-100, providing sufficient training iterations without excessive computational cost.

**Data augmentation:** We employ the standard CIFAR augmentation recipe: random crop with 4-pixel padding and horizontal flip, combined with per-channel normalization. Notably, we do *not* use stronger augmentations such as CutMix, Mixup, or AutoAugment, ensuring the baseline does not gain unfair advantages from advanced data augmentation techniques.

### C.5.4 COATNET-CIFAR RESULTS

Table 26 presents the complete results across all five random seeds (42–46) for each configuration.

Table 17: Optimal alpha values for Random Seed 43

| Architecture | 1X | 10X | 20X |
|---|---|---|---|
| ResNet-18 | 0.0042 | 0.5623 | 0.2371 |
| ViT-Small | 31.6228 | 0.0316 | 0.0056 |
| ResNet-50 | 5.6234 | 0.0750 | 0.1000 |
| ViT-B/16 | 23.7137 | 0.0316 | 0.0316 |

Table 18: Optimal alpha values for Random Seed 44

| Architecture | 1X | 10X | 20X |
|---|---|---|---|
| ResNet-18 | 0.0178 | 0.2371 | 0.0178 |
| ViT-Small | 31.6228 | 0.0133 | 0.0042 |
| ResNet-50 | 0.5623 | 0.0042 | 0.0316 |
| ViT-B/16 | 31.6228 | 0.0237 | 0.0178 |

## D ATTENTION PATTERN EVOLUTION ANALYSIS

### D.1 DETAILED ATTENTION VISUALIZATION

Figure 6 visualizes the evolution of self-attention maps throughout the training trajectory.

### D.2 QUANTITATIVE ATTENTION ANALYSIS

We measured attention pattern quality using several metrics:

Key findings:

- **Peak Strength**: Asymmetric training produces more diffuse attention patterns (lower peak values)
- **Map Entropy**: Higher entropy indicates more distributed attention across spatial locations
- **Sparsity**: Lower Gini coefficient suggests more egalitarian attention distribution
- **Object Coverage**: Asymmetric models achieve near-optimal object coverage much earlier (Epoch 20 vs 100)

### D.3 LAYER-WISE ATTENTION DEVELOPMENT

The layer-wise analysis reveals that asymmetric training guides the development of hierarchical attention patterns: - **Early layers (L1)**: Both variants show similar low-level feature attention - **Middle layers (L3)**: Asymmetric variant begins showing more structured patterns - **Late layers (L6)**: Clear differentiation—asymmetric model develops coherent object-level attention while plain model remains diffuse

## E STATISTICAL VALIDATION

All reported results were validated using appropriate statistical tests. For performance comparisons, we used paired t-tests with Bonferroni correction for multiple comparisons. Effect sizes were calculated using Cohen's d, with the following interpretations: small (0.2), medium (0.5), large (0.8).

All main results show statistical significance ($p < 0.001$) with large effect sizes, confirming the reliability of our findings.

To investigate the formation process of the final attention patterns, we visualized the evolution of attention across different training stages (e.g., 20, 60, 100 epochs), as shown in Figure 6. We observe

Table 19: Optimal alpha values for Random Seed 45

| Architecture | 1X | 10X | 20X |
|---|---|---|---|
| ResNet-18 | 0.0032 | 0.0422 | 0.1778 |
| ViT-Small | 31.6228 | 0.0133 | 0.0422 |
| ResNet-50 | 17.7828 | 0.0042 | 0.0237 |
| ViT-B/16 | 31.6228 | 0.0237 | 0.0237 |

Table 20: Training hyperparameters for ImageNet-1k experiments

| Parameter | ResNet-18 | ViT-Small | ResNet-50 | ViT-B/16 |
|---|---|---|---|---|
| Optimizer | SGD | AdamW | SGD | AdamW |
| Learning Rate | 0.2 | 1e-3 | 0.2 | 1e-3 |
| Momentum | 0.9 | - | 0.9 | - |
| Weight Decay | 1e-4 | 0.1 | 1e-4 | 0.3 |
| Batch Size | 512 | 256 | 512 | 256 |
| Total Epochs | 90 | 150 | 90 | 150 |
| LR Schedule | StepLR | Cosine | StepLR | Cosine |
| Step Epochs | 30/60/80 | - | 30/60/80 | - |
| Warmup | - | 5% | - | 5% |
| Min LR | - | 1e-5 | - | 1e-5 |

*Note: Learning rate for ResNets uses linear batch scaling from base 0.1. No heavy regularizers (Mixup, CutMix) used. Gradient clipping and zero-initialized heads applied uniformly.*

that the attention patterns of the Asymmetric model gradually become more holistic and comprehensive as training progresses. In contrast, the attention of the Plain model saturates earlier and consistently focuses more on local textures.

### E.1 PROCESS LEVEL: LEARNING TRAJECTORIES

This microscopic gradient behavior directly translates into dramatically different macroscopic learning dynamics, as evidenced by our analysis across Figure 8 and Tables 31-32. For CNN, the gradient conflict drives a catastrophic optimization collapse—the model converges prematurely in just 5 epochs to a inferior solution with 71% performance degradation. More tellingly, the generalization gap becomes negative by epoch 50 (-0.0242), indicating the model performs better on validation than training data—a clear symptom of learning failure. In contrast, ViT's constructive gradient synergy guides the optimization along a more exploratory but ultimately superior trajectory, requiring 11 additional epochs but achieving both 30.6% higher validation accuracy and 47% better generalization (gap reduction from 0.5543 to 0.2934). This demonstrates that beneficial gradient alignment not only improves final performance but fundamentally enhances the learning process itself.

Table 21: Detailed Architectural Configurations for ImageNet-1k Experiments (Table 2). We summarize the backbone specifications and the exact locations of auxiliary attachment points. For ResNet, attachments are made at the feature maps of each stage prior to global pooling. For ViT, attachments are made to the CLS token at evenly spaced block intervals. The redundancy level $N$ (number of branches per point) is a hyperparameter (e.g., $N = 20$).

| Architecture | Backbone Specification | Aux. Attach Points | Feature Dim ($d_i$) |
|---|---|---|---|
| **ResNet-18** | 4-stage CNN (BasicBlock) GAP + FC(512→1000) | End of Stages 1, 2, 3, 4 (before pooling) | 64, 128, 256, 512 |
| **ResNet-50** | 4-stage CNN (Bottleneck) GAP + FC(2048→1000) | End of Stages 1, 2, 3, 4 (before pooling) | 256, 512, 1024, 2048 |
| **ViT-Small** | 12 blocks, $D$=384, 6 heads CLS head 384→1000 | CLS token after Blocks 3, 6, 9 | 384 (at all points) |
| **ViT-B/16** | 12 blocks, $D$=768, 12 heads CLS head 768→1000 | CLS token after Blocks 3, 6, 9 | 768 (at all points) |

**Auxiliary Head Configuration (Shared):**
At each attachment point, we instantiate $N$ parallel branches. The first $N - 1$ branches are **static** (weights orthogonally initialized and frozen/non-trainable), while the $N$-th branch is **active** (fully trainable). All branches project features to 1000-dim logits for Sigmoid/BCE supervision.

Table 22: Two-stage hyperparameter search strategy for $\alpha$ in Asymmetric architecture on ImageNet-1k

| Stage | Search Range | Candidate Values | Strategy |
|---|---|---|---|
| Stage 1 (Coarse) | $[10^{-2}, 10^1]$ $[0.01, 10.0]$ | $\alpha = 0.01$ $\alpha = 0.056$ $\alpha = 0.316$ $\alpha = 1.778$ $\alpha = 10.0$ | `np.logspace(-2, 1, 5)` Log-uniform sampling 5 candidates |
| Stage 2 (Fine) | $[10^{\log_{\text{best}} -0.5},$ $10^{\log_{\text{best}} +0.5}]$ | $\alpha_1$ $\alpha_2$ $\alpha_3$ $\alpha_4$ $\alpha_{\text{best}}$ (center) $\alpha_6$ $\alpha_7$ $\alpha_8$ $\alpha_9$ | `np.logspace(` $\log_{\text{best}} -0.5,$ $\log_{\text{best}} +0.5, 9)$ Local refinement around best value 9 candidates |

Table 23: An Example of Stage 2 candidates when Stage 1 optimal is $\alpha = 0.316$

| Index | $\alpha$ Value |
|---|---|
| 1 | 0.100 |
| 2 | 0.133 |
| 3 | 0.178 |
| 4 | 0.237 |
| 5 | 0.316 (optimal from Stage 1) |
| 6 | 0.422 |
| 7 | 0.562 |
| 8 | 0.750 |
| 9 | 1.000 |

Table 24: Hyperparameter configuration for CoAtNet baseline experiments on CIFAR-100.

| Category | Parameter | Value |
|---|---|---|
| Dataset | Dataset | CIFAR-100 |
| | Split | Official train/test |
| Data Preprocessing | Input resolution | $32 \times 32$ |
| | Data augmentation | Random crop (4-pixel padding) |
| | | + Random horizontal flip |
| | Normalization | Per-channel mean/std normalization |
| Optimizer | Type | AdamW |
| | Base learning rate | $1 \times 10^{-3}$ |
| | Weight decay | 0.05 |
| | $\beta_1, \beta_2$ | 0.9, 0.999 |
| Learning Rate Schedule | Warmup epochs | 5 |
| | Schedule type | Cosine decay |
| | Minimum learning rate | $1 \times 10^{-6}$ |
| Training | Total epochs | 200 |
| | Batch size | 128 |
| Regularization | Drop path rate | 0.1 |
| | Label smoothing | 0.1 |
| Model | Architecture | CoAtNet-CIFAR (fixed structure) |
| | Auxiliary head | None (aux_stage = None, $\alpha = 0$) |

Table 25: Architecture configuration of CoAtNet-CIFAR baseline model.

| Stage | Block Type | Channels | Resolution | # Blocks | Operation |
|---|---|---|---|---|---|
| S0 | Conv Stem | 64 | $32 \times 32 \rightarrow 32 \times 32$ | 1 | Conv $3 \times 3$ |
| S1 | MBConv | 96 | $32 \times 32 \rightarrow 16 \times 16$ | 2 | + Downsample |
| S2 | MBConv | 128 | $16 \times 16 \rightarrow 8 \times 8$ | 2 | + Downsample |
| S3 | Transformer | 256 | $8 \times 8 \rightarrow 8 \times 8$ | 3 | Self-Attention |
| S4 | Transformer | 384 | $8 \times 8 \rightarrow 4 \times 4$ | 2 | + Downsample |

Table 26: Detailed test accuracy results across five random seeds for all configurations (CoAtNet-CIFAR).

| Redundancy | Stage | Alpha | Seed 42 | Seed 43 | Seed 44 | Seed 45 | Seed 46 | Mean | Std |
|---|---|---|---|---|---|---|---|---|---|
| 1× | S2 | 4.642 | 75.63 | 75.34 | 75.94 | 75.18 | 74.21 | 75.26 | 0.66 |
| | S3 | 21.544 | 75.94 | 76.14 | 75.64 | 75.60 | 76.21 | 75.91 | 0.28 |
| 20× | S2 | 0.215 | 75.96 | 75.90 | 75.79 | 76.05 | 75.70 | 75.88 | 0.14 |
| | S3 | 4.642 | 76.85 | 76.43 | 77.07 | 76.31 | 77.14 | 76.76 | 0.36 |
| 100× | S2 | 0.1 | 75.34 | 75.38 | 76.32 | 75.54 | 75.39 | 75.59 | 0.41 |
| | S3 | 4.642 | 77.41 | 77.32 | 76.95 | 76.74 | 76.96 | 77.08 | 0.26 |
| 300× | S2 | 0.1 | 74.41 | 73.97 | 74.42 | 75.20 | 74.70 | 74.54 | 0.49 |
| | S3 | 0.1 | 76.72 | 77.22 | 76.96 | 76.76 | 76.44 | 76.82 | 0.30 |

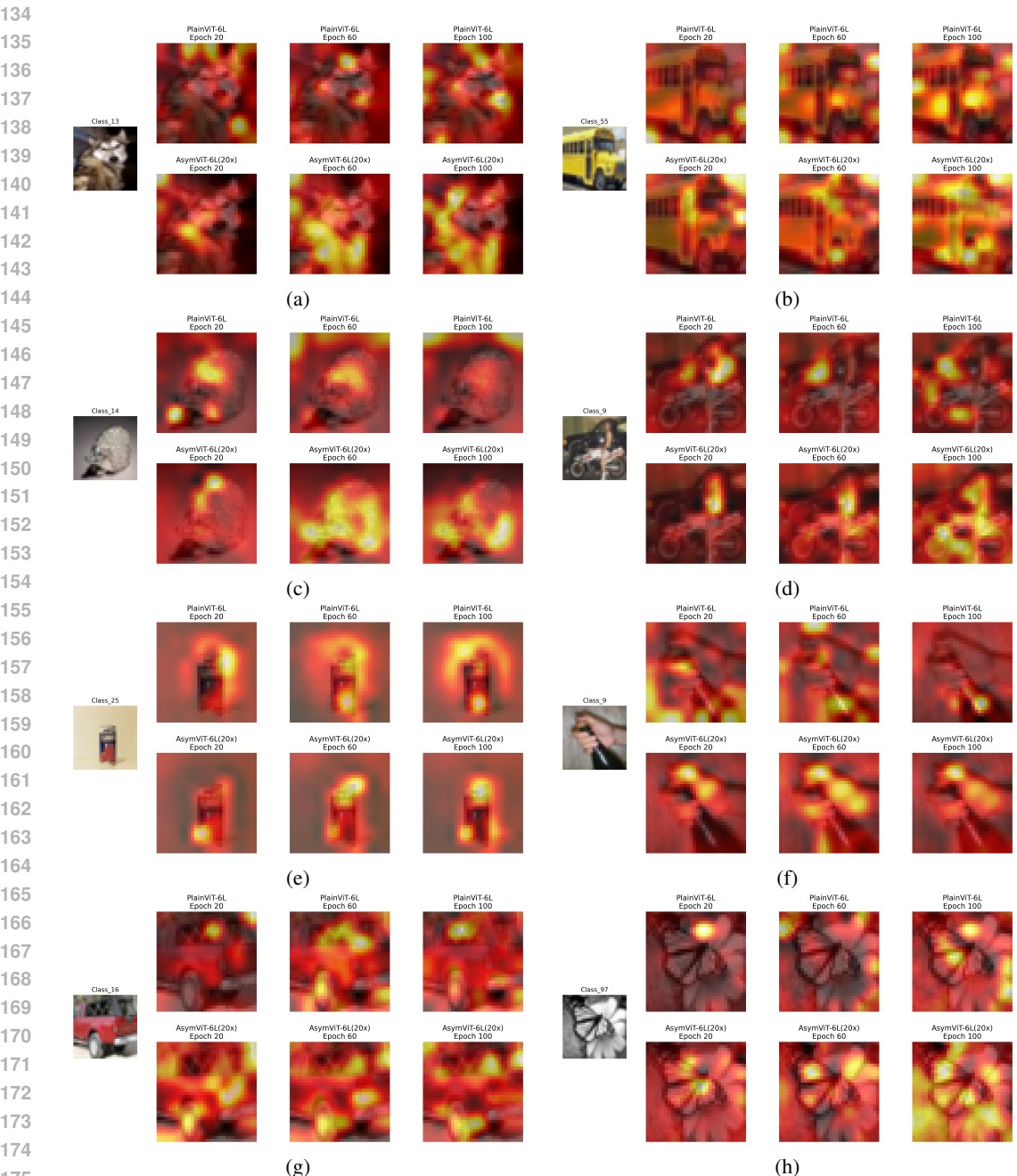

Figure 6: Comprehensive attention pattern evolution for ViT-6L on CIFAR-100.

Table 27: Attention pattern quality metrics across training epochs

| Metric | Plain ViT-6L | | | Asymmetric ViT-6L | | |
|---|---|---|---|---|---|---|
| | Epoch 20 | Epoch 60 | Epoch 100 | Epoch 20 | Epoch 60 | Epoch 100 |
| Peak Strength | 0.092 | 0.063 | 0.051 | 0.053 | 0.049 | 0.045 |
| Map Entropy | 3.73 | 3.89 | 3.97 | 4.02 | 4.03 | 4.05 |
| Sparsity (Gini) | 0.471 | 0.415 | 0.374 | 0.254 | 0.287 | 0.302 |
| Object Coverage | 0.58 | 0.70 | 0.84 | 0.96 | 0.95 | 0.94 |

Table 28: Gradient Conflict Analysis. Cosine similarity analysis between main (softmax) and auxiliary (sigmoid) gradients during training. ViT-6L shows consistent positive similarity (synergy), while CNN exhibits strong negative similarity (conflict), and MLP demonstrates near-orthogonal gradients with slight conflict tendency.

| Architecture | Final Similarity | Average Similarity | Min Similarity | Max Similarity |
|---|---|---|---|---|
| MLP | -0.0309 | -0.0801 | -0.3422 | 0.0701 |
| CNN | -0.1926 | -0.2574 | -0.8210 | 0.0464 |
| ViT-6L | 0.2631 | 0.1870 | -0.1845 | 0.3654 |

Table 29: To explore the impact of different hyperparameter search strategies, we conducted a supplementary experiment for the $N_{\text{branches}} = 3$ configuration, where hyperparameters (learning rate and weight decay) were independently optimized for both baseline and asymmetric models. The results are shown in the table, where "all active" indicates that all three auxiliary branches at each connection point are trainable, while "one active" means only one auxiliary branch per connection point is trainable. Although this "dual optimization" strategy can yield benefits in certain cases, we consistently adopted the "Pragmatic Gold Standard" strategy throughout the main text to isolate the pure effect of our paradigm.

| $N_{branches}$ | Architecture | Plain | Asymmetric | Improvement | p-value |
|---|---|---|---|---|---|
| 3×branches (all active) | MLP | 24.6 ± 0.3 | 20.7 ± 0.5 | -3.9 | 0.0000 |
| | CNN | 39.9 ± 0.6 | 36.2 ± 0.6 | -3.7 | 0.0000 |
| | ViT-6L | 36.2 ± 0.6 | 42.1 ± 2.7 | +5.9 | 0.0002 |
| 3×branches (one active) | MLP | 24.6 ± 0.3 | 23.7 ± 0.6 | -0.9 | 0.0008 |
| | CNN | 40.0 ± 0.8 | 33.9 ± 0.9 | -6.1 | 0.0000 |
| | ViT-6L | 36.5 ± 0.7 | 44.0 ± 5.9 | +7.5 | 0.0045 |

Table 30: Architecture performance comparison with asymmetric training on CIFAR-10. Results show differential architectural responses to auxiliary supervision, with statistical significance assessed using two-tailed paired t-tests across 10 independent runs.

| Architecture | Baseline | Asymmetric | Improvement | p-value | Significant |
|---|---|---|---|---|---|
| MLP | 49.5 ± 0.3 | 50.6 ± 0.4 | +1.1 | 0.0001 | Yes |
| CNN | 77.0 ± 0.6 | 77.2 ± 1.0 | +0.2 | 0.5371 | No |
| ViT-6L | 62.5 ± 0.7 | 63.6 ± 0.4 | +1.1 | 0.0068 | Yes |

Table 31: Architecture-dependent convergence patterns and performance outcomes.

| | Convergence (Epoch) | | Final Val Acc | |
|---|---|---|---|---|
| Architecture | Plain | Asymmetric | Plain | Asymmetric |
| MLP | 15 | 21 | 0.2220 | 0.2305 |
| CNN | 20 | 5 | 0.4034 | 0.1171 |
| ViT-6L | 19 | 30 | 0.3498 | 0.4568 |

Table 32: Generalization Gap Evolution Across Training Phases

| | Early(epoch10) | | Middle(epoch50) | | Late(epoch100) | |
|---|---|---|---|---|---|---|
| Architecture | Plain | Asymmetric | Plain | Asymmetric | Plain | Asymmetric |
| MLP | 0.0098 | 0.0049 | 0.1039 | 0.0566 | 0.2234 | 0.1225 |
| CNN | 0.0073 | 0.0002 | 0.4665 | -0.0242 | 0.5253 | -0.0365 |
| ViT-6L | 0.0334 | -0.0270 | 0.4717 | 0.0838 | 0.5543 | 0.2934 |

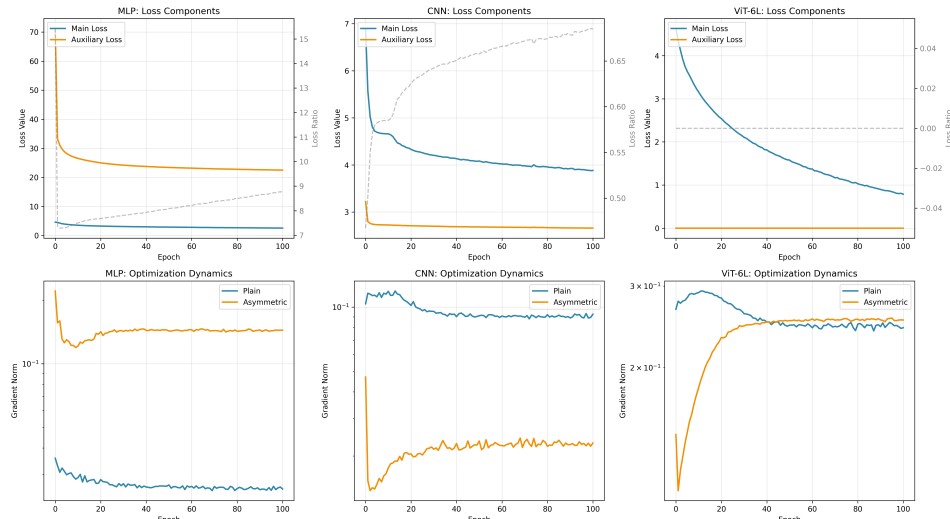

Figure 7: Training dynamics comparison across architectures showing loss components and optimization trajectories. **Top row**: Evolution of main loss (softmax) and auxiliary loss (sigmoid) during training. **Bottom row**: Gradient norm dynamics for plain and asymmetric variants. Asymmetric training exhibits architecture-specific patterns: MLP shows stable auxiliary loss with reduced gradient norms, CNN demonstrates auxiliary loss divergence with increased gradient instability, while ViT-6L displays rapid auxiliary loss convergence with improved optimization stability.

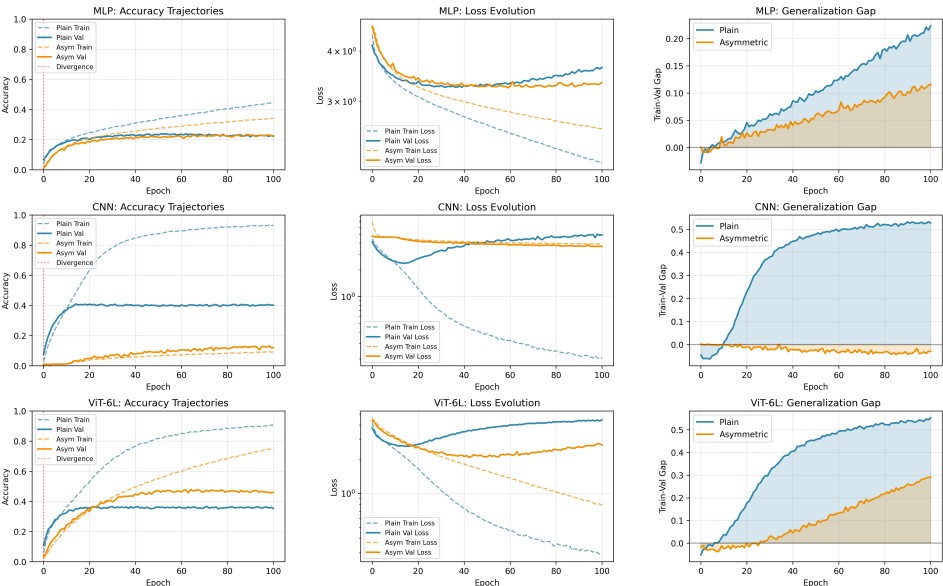

Figure 8: Architecture-specific learning dynamics reveal the Principle of Architectural Resonance. Comprehensive learning trajectories comparing Plain (baseline) and Asymmetric training across three architectures on CIFAR-100. Left column shows accuracy evolution (dashed: training, solid: validation); middle column displays loss curves; right column presents generalization gaps (train-val accuracy difference). CNN exhibits catastrophic degradation with massive overfitting under asymmetric training. MLP demonstrates effective regularization with reduced generalization gap but limited accuracy gains. ViT achieves substantial performance improvements with superior generalization. The divergence points (red vertical lines) mark early onset of architecture-dependent responses to heterogeneous supervision, empirically validating our core hypothesis that auxiliary signal efficacy depends fundamentally on architectural inductive biases.

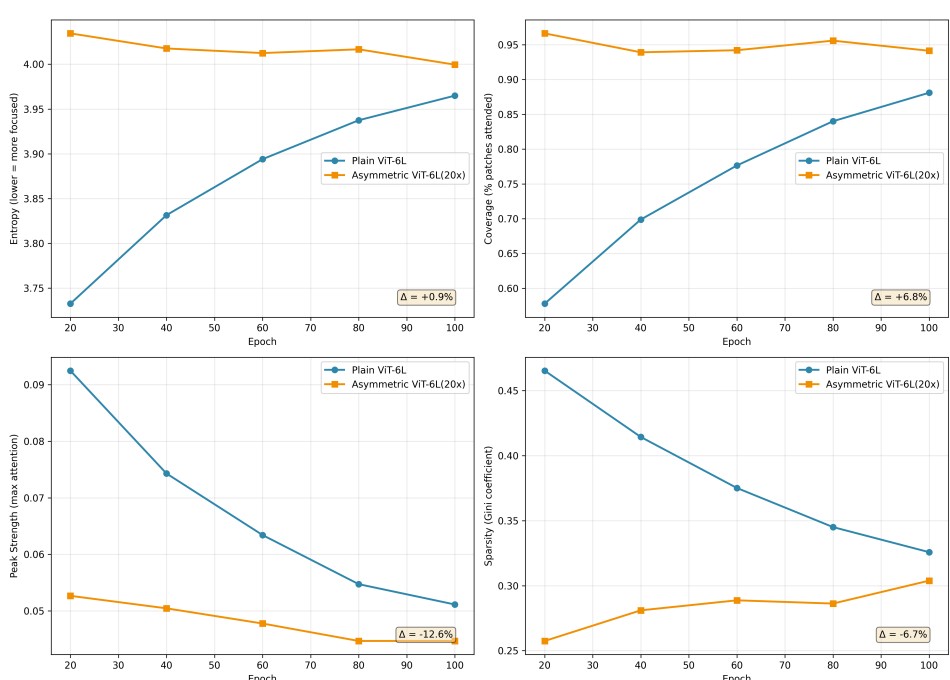

Figure 9: ViT-6L attention mechanism analysis comparing plain and asymmetric training. **Top left**: Entropy evolution showing asymmetric training maintains higher attention diversity. **Top right**: Coverage percentage demonstrating improved spatial attention coverage (+6.8%). **Bottom left**: Peak strength indicating more focused attention patterns (-12.6%). **Bottom right**: Sparsity coefficient revealing attention distribution characteristics (-6.7%). Results suggest asymmetric training promotes more comprehensive yet focused attention patterns.

