# OpenReview forum: "Asymmetric Training with Heterogeneous Losses: A Probe into Architectural Resonance"
_ICLR.cc/2026/Conference — Submitted to ICLR 2026_

### Official Review · Reviewer_1ZLR · 2025-10-14

**Soundness:** 2
**Presentation:** 1
**Contribution:** 1
**Rating:** 2
**Confidence:** 4

**Summary:**

This paper designs an asymmetric training paradigm that temporarily introduces noncompetitive class supervision (sigmoid loss) into a network optimized with a competitive softmax objective. This mechanism is implemented via orthogonally initialized auxiliary paths, which are controlled by a scalar coefficient α and activated only during training. This controllable form of temporary topological redundancy provides an ideal probe for studying target interactions.

**Strengths:**

The PPL is clear. The content is logical and there are no issues. The asymmetrical method is quite good in high level.

**Weaknesses:**

The study's core mechanism design, the "single activation" strategy, suffers from serious logical flaws and misattributes its role. The authors claim that the sole purpose of adding a large number of static auxiliary branches that are never activated or trained is to smooth the initial loss landscape. This claim is highly unusual and unconvincing. Attributing the observed stark differences in performance (ViT benefits, CNN suffers) solely to the geometric effects of this static topology on the initial landscape is a significant leap in logic. A more straightforward and likely explanation is that this peculiar structure introduces an unexplored implicit regularization, or that different architectures simply have vastly different sensitivities to the hyperparameters (particularly the value of α) of this structure and heterogeneous loss combination. The paper fails to provide strong evidence to rule out these simpler explanations, placing its core conclusion, "architectural resonance," on shaky ground.

The paper overgeneralizes the phenomena observed in a highly controlled and simplified experimental setting to a universal "architectural resonance principle," significantly undermining the persuasiveness of its conclusion. All experiments were conducted on small datasets (CIFAR-10/100) and specially designed lightweight models (e.g., ViT and CNN with only 6 layers). The inductive biases and optimization dynamics of these "toy models" are likely very different from those of deeper and more complex standard models (e.g., ResNet-50 or ViT-B) pre-trained on large-scale datasets like ImageNet. Promoting a regularity discovered in such a limited setting to a "fundamental principle" is both hasty and imprecise. Without validation on larger and more representative models, these findings should be viewed as an interesting preliminary case study rather than a generalizable principle that can guide practice.

The study's experimental design failed to effectively decouple and control variables, resulting in flawed causal arguments. The paper claims to follow a "practical gold standard" hyperparameter search strategy, namely, fixing the optimal learning rate and weight decay for the baseline model and then searching for the optimal auxiliary loss weight α only for the asymmetric model. This approach is methodologically problematic. Introducing a completely new loss term that potentially conflicts with the primary task significantly alters the optimization dynamics of the entire model, almost inevitably leading to changes in the optimal learning rate and weight decay. Therefore, the observed performance differences are likely not a pure manifestation of "architectural resonance," but rather the result of one architecture (such as a CNN) falling into severe suboptimal hyperparameter settings under the new training paradigm, while another architecture (ViT) happens to be more tolerant of this mismatch. This makes performance comparisons between different architectures unfair and weakens the persuasiveness of the conclusions drawn.

**Questions:**

The paper directly equates the phenomenon of negative gradient cosine similarity with a "destructive conflict" that is harmful to CNNs. However, in multi-task learning and regularization theory, conflict in gradient direction is sometimes viewed as a beneficial constraint, forcing the model to learn more general and robust feature representations that balance the needs of different tasks. Why do you a priori define this conflict as a purely negative effect, rather than a potentially beneficial regularization pressure that this particular CNN architecture cannot effectively exploit due to its capacity or inductive bias? How can we be sure that the observed performance degradation is the inevitable result of a failure of "architectural resonance" rather than a manifestation of an inability to converge to a better generalization point from this gradient "tug-of-war" due to insufficient model capacity, poor optimizer selection, or hyperparameter mismatch?

The principle of "architectural resonance" you propose is currently presented as a binary opposition: ViT is "synergistic" and CNN is "destructive." This division seems overly simplistic. Modern neural network architectures, such as ConvNeXt, have begun to combine the locality of CNNs with the global dependency modeling capabilities of ViT. According to your theory, how will these hybrid architectures respond to heterogeneous losses? Will they exhibit synergy, destructiveness, or something in between? More importantly, if this principle holds true, what specific implications does it have for future model design? Does it mean that we should completely avoid using non-competitive auxiliary losses in CNNs, or should we design new auxiliary signals that resonate with the CNN's local inductive biases?

The core of your experimental design—the "single activation" strategy—is quite unique. You argue that its purpose is to isolate the variable of "initial loss landscape geometry." However, this raises a key question: How do you prove that the full impact of these N-1 statically, orthogonally initialized branches on the training dynamics can truly be reduced to simply "smoothing the initial landscape"? Have you conducted the necessary control experiments to confirm this? For example, by introducing an equal number of static parameters connected in a different way (not in the auxiliary classification head) to verify whether the so-called "resonance" effect is unique to this specific topology? Without such verification, we cannot rule out that this phenomenon is simply a side effect of increasing the number of parameters, changing the weight initialization distribution, or introducing some unknown form of regularization.

---

> ### Author Response · Authors · 2025-11-22
> **Response to Reviewer 1ZLR: ImageNet Validation & Rigorous Controls**
>
> We sincerely thank the reviewer for the critical assessment. Addressing your concerns on generalization and confounds, we conducted rigorous controls and ImageNet validation, which significantly strengthened our work.
>
> **Q1. "Toy Models" & Generalization (External Validity)**
>
> The reviewer rightly questioned whether our findings on CIFAR transfer to standard scales. We addressed this by conducting ImageNet-1k experiments with ResNet-18/50 and ViT-Small/B-16 (4 seeds).
> 1. Results: Validation of Architectural Resonance
>
> As shown in Table A, the phenomenon is not limited to toy models.
> * ViTs: Show consistent, statistically significant improvements (+1.77% to +3.37%), with gains increasing on larger models (ViT-B/16).
> * ResNets: Show no gain or slight degradation, confirming that the "Architecture-Objective Mismatch" holds for large-scale CNNs (though likely buffered by residual connections).
>
> Table A. ImageNet-1k
> Architecture | Baseline | 1X | 10X | 20X
> -------------|----------|-----|------|-----
> ResNet-18 | 68.23±0.20 | 68.33±0.10 (+0.15%, p=0.33) | 68.29±0.10 (+0.09%, p=0.51) | 68.37±0.13 (+0.21%, p=0.21)
> ResNet-50 | 73.83±0.09 | 73.55±0.17 (-0.38%, p=0.01) | 73.75±0.11 (-0.11%, p=0.41) | 73.76±0.21 (-0.09%, p=0.43)
> ViT-Small | 69.13±0.14 | 70.16±0.09 (+1.49%, p=0.001) | 70.27±0.18 (+1.65%, p=0.003) | 70.35±0.29 (+1.77%, p=0.004)
> ViT-B/16 | 66.75±0.13 | 68.34±0.64 (+2.38%, p=0.02) | 68.83±0.27 (+3.11%, p<0.001) | 69.00±0.19 (**+3.37%**, p<0.001)
>
> 2. Experimental Rigor
>
> To ensure these are not artifacts of hyperparameter hacking, we followed Pragmatic Gold Standard protocols (SGD for ResNet, AdamW for ViT (batch size 256 on single GPU), standard augmentation, no heavy regularization). We tuned $\alpha$ on a 10% held-out training subset, ensuring the validation set remained strictly unseen during search.
>
> **Q2. Is this just "Implicit Regularization" or "Initialization Artifacts"?**
>
> To rule out initialization artifacts or simple parameter expansion, we conducted specific controls.
>
> 1. Control: Random vs. Orthogonal Initialization
>
> We ran experiments with randomly initialized auxiliary branches (10 seeds).
>
> Result (Table B): Random initialization fails. CNN consistently degrades (-20.0% to -23.9%, p<0.0001), and ViT-6L shows no benefit or degradation. In contrast, our Orthogonal Initialization produces consistent landscape smoothing and performance gains. These findings indicate that the benefit is tied to the specific structural design of our method, not just adding random noise/parameters.
>
> Table B. Random Initialization Control
> N | MLP Baseline→Result | CNN Baseline→Result | ViT-6L Baseline→Result
> -------|---------------------|---------------------|--------------------
> 1X | 0.233±0.003→0.243±0.004(+4.3%,p=0.0001) | 0.395±0.009→0.316±0.007(-20.0%,p<0.0001) | 0.360±0.010→0.363±0.009(+0.8%,p=0.64)
> 10X | 0.231±0.003→0.228±0.003(-1.3%,p=0.068) | 0.402±0.007→0.306±0.010(-23.9%,p<0.0001) | 0.361±0.010→0.356±0.011(-1.4%,p=0.38)
> 20X | 0.233±0.003→0.230±0.002(-1.3%,p=0.085) | 0.399±0.006→0.308±0.008(-22.8%,p<0.0001) | 0.358±0.008→0.318±0.011(-11.2%,p=0.0001)
>
>
> 2. Control: Capacity & Heterogeneity
>
> To further address the "Implicit Regularization" concern, we conducted two additional controls:
> * Capacity Control: Adding branches excluded from the total loss yielded results indistinguishable from baseline, showing that parameter count alone does not explain gains.
> * Heterogeneity Control: Using Softmax (homogeneous) instead of Sigmoid (heterogeneous) on auxiliary branches leads to performance drops (e.g., -5.3% on ViT-6L), confirming that improvements arise specifically from heterogeneous resonance rather than generic deep supervision. See Response to Reviewer 1whc (Tables B & C) for details.
>
> **Q3. Hyperparameter Search Strategy**
>
> Our "Pragmatic Gold Standard" strategy provides a more realistic evaluation scenario. In practice, when researchers apply Asymmetric Training Paradigm to existing models, they typically keep the proven base hyperparameters and only tune the auxiliary weight $\alpha$ as a plug-and-play component. A full re-optimization for each architecture would introduce confounding factors and make it impossible to isolate the pure effect of auxiliary supervision.
>
> Taken together, these controls and validations clarify that our findings are neither “toy model” artifacts nor simple implicit regularization. The phenomenon of Architectural Resonance is consistently observed on ImageNet, specific to orthogonal/heterogeneous auxiliary design (vs. random or homogeneous controls), and driven by architecture‑objective compatibility. While our evidence is limited to tested architectures and datasets, the consistency across scales suggests broader applicability. We present this principle as an initial step toward deeper understanding of architecture–objective interactions, suggesting specifically that ViTs require heterogeneous regularization to unlock capacity, whereas CNNs strictly demand objective alignment.

---

> > ### Comment · Reviewer_1ZLR · 2025-11-22
> >
> > I will stay my score due to insufficient explanation.

---

> > > ### Author Response · Authors · 2025-11-22
> > > **Reply to Reviewer 1ZLR**
> > >
> > > Thank you for the clarification.
> > >
> > > We fully respect your decision and appreciate the time and care you have devoted to reviewing our submission. Your feedback has been helpful, and we will take it into consideration for improving the manuscript.
> > >
> > > Thank you again for your thoughtful review.

---

> > > > ### Comment · Reviewer_s9FL · 2025-11-26
> > > >
> > > > Is there a possibility to get the author's response on the following question/concern by reviewer 1ZLR?
> > > >
> > > > > The principle of "architectural resonance" you propose is currently presented as a binary opposition: ViT is "synergistic" and CNN is "destructive." This division seems overly simplistic.....

---

> > > > > ### Author Response · Authors · 2025-11-27
> > > > > **Response to Reviewer s9FL (Clarification on Architectural Resonance Spectrum)**
> > > > >
> > > > > We sincerely appreciate your thoughtful comment and the opportunity to clarify this point. We agree that "Architectural Resonance" is best understood as a working hypothesis regarding the interaction between objectives and inductive biases, manifesting along a spectrum.
> > > > >
> > > > > **1. Empirical Spectrum & Gradient Interaction**
> > > > >
> > > > > Our existing analysis reveals that the spatially-agnostic (dense) auxiliary projection induces a continuous spectrum of alignment: positive gradient alignment in ViTs (**+0.19**, Resonance), consistently negative in CNNs (**-0.26**, Conflict), and near-zero in MLPs (**-0.08**, Noise-like). This gradient pattern supports our hypothesis.
> > > > >
> > > > > **2. Buffered Conflicts in CNNs**
> > > > >
> > > > > Under this dense probe topology, ViTs (global representations) exhibit consistent synergy. CNNs (local priors) clash with the spatially-agnostic probe, causing degradation at simpler scales (CIFAR: **-8.7%**). However, at larger scales (ImageNet), architectural resilience mechanisms (residual connections, normalization) effectively buffer these conflicts, explaining the near-neutral outcomes for ResNet-18/50 compared to the collapse of plain CNNs.
> > > > >
> > > > > **3. Hypotheses on Hybrid Architectures**
> > > > >
> > > > > For Hybrid architectures (e.g., ConvNeXt, CoAtNet), our hypothesis implies a stage-dependent response rather than a uniform one: we expect conflict in early convolutional stages (due to probe-bias mismatch) but potential synergy in later global/attention stages.
> > > > >
> > > > > **4. Design Implications**
> > > > >
> > > > > This leads to a clear design guideline: rather than discarding auxiliary objectives for CNN-based models, future work could explore locality-aware auxiliary signals (e.g., convolutional heads) that align with the backbone's inductive bias while retaining the optimization benefits of redundancy and heterogeneity.
> > > > >
> > > > >
> > > > > Our contribution is not to claim a universal law, but to provide empirical evidence supporting the Architectural Resonance Hypothesis under controlled probes, and to highlight its potential implications for the bias-aligned design of heterogeneous auxiliary signals in future architectures.

---

> ### Author Response · Authors · 2025-12-03
> **Update: Targeted Verification on Hybrid Architecture (CoAtNet)**
>
> To empirically validate our "Stage-Dependent" hypothesis (addressing concerns regarding "binary opposition" from Reviewer s9FL and Reviewer 1ZLR), we conducted a targeted experiment on a CIFAR-100 adapted CoAtNet-0 backbone.
>
> Setup: We applied our asymmetric probe separately to the Convolutional Stage (S2) and the Transformer Stage (S3) to test their differential responses.
>
> **Table: Stage-Dependent Response on CIFAR-100 adapted CoAtNet-0.** Comparison of applying auxiliary supervision to CNN (S2) vs. Transformer (S3) stages across varying redundancy levels ($N_{branches}$).
>
> | $N_{branches}$ | Baseline | S2 (CNN Stage) | S3 (Transformer Stage) | Phenomenon |
> | :--- | :--- | :--- | :--- | :--- |
> | -- | 75.34±0.20 | -- | -- | -- |
> | 1× | | 75.26±0.59 (-0.08%, n.s.) | 75.91±0.28 (+0.57%, **) | ViT starts gaining |
> | 20× | | 75.88±0.14 (+0.54%, **) | 76.76±0.37 (+1.42%, ***) | Strong Synergy (S3) |
> | 100× | | 75.59±0.44 (+0.25%, n.s.) | **77.08±0.27** (**+1.74%**, ***) | **Peak Resonance (S3)** |
> | 300× | | **74.54±0.48** (**-0.80%**, *) | 76.82±0.30 (+1.48%, ***) | **Conflict (S2)** |
>
> **Statistical significance:** * p < 0.05, ** p < 0.01, *** p < 0.001.
>
> Key Finding: While Transformer stages (S3) exhibit consistent, statistically significant resonance,
> CNN stages (S2) show limited capacity for auxiliary supervision, exhibiting a statistically significant tendency toward conflict (**-0.80%**) at high redundancy.
>
> The results reveal an observable divergence: while Transformer stages benefit consistently from heterogeneity (Synergy), Convolutional stages exhibit a tendency toward conflict at high redundancy. These findings provide targeted empirical support for our Architectural Resonance Hypothesis, specifically validating its stage-dependent implications in hybrid architectures.

---

### Official Review · Reviewer_s9FL · 2025-10-31

**Soundness:** 2
**Presentation:** 2
**Contribution:** 3
**Rating:** 4
**Confidence:** 2

**Summary:**

The paper studies architecture-objective interactions by attaching training-only heterogeneous auxiliary branches (sigmoid BCE) to models whose main head uses softmax CE. Only one main branch is active during training; branches are removed at inference. The claim is a Principle of Architectural Resonance: ViTs benefit, CNNs degrade, MLPs mixed, explained via gradient cosine similarity between main and auxiliary losses. The paper was difficult to follow in several places, and some implementation details were not clear to me.

**Strengths:**

- Clear experimental knob: vary auxiliary-loss weight and number of branches, then measure gradient cosine similarity, not only accuracy.
- No inference overhead since auxiliary branches are removed.
- ViT gains on CIFAR-100 appear consistent with the reported alignment signal.

**Weaknesses:**

- Inactive branches vs. claimed smoothing: The paper states that only one branch is active in forward and backward passes, while the rest never participate. If so, it is unclear how additional inactive branches can smooth the initial loss landscape. This requires a precise computational explanation.

- External validity: Results are limited to small CIFAR models. No ImageNet-scale or strong pretrained backbones, so the generality of the claimed principle is uncertain.

- Overall, the paper was hard to understand, reducing confidence in the conclusions.

**Questions:**

- Do the inactive branches contribute in any way to the forward graph or loss computation? If not, how do they alter loss-landscape statistics? Please specify the exact computation path and any regularizers.
- Can the principle be validated on at least one ImageNet-scale experiment or a pretrained ViT-B and a stronger CNN with normalization and residuals?
- How does this compare against deep supervision baselines and gradient-conflict methods that explicitly manage alignment?

---

> ### Author Response · Authors · 2025-11-22
> **Response to Reviewer s9FL: Mechanism & ImageNet Results**
>
> We sincerely thank the reviewer for constructive feedback, which helped refine our terminology and strengthen evidence.
>
> **Q1. The Role of "Inactive" Branches and Loss Landscape**
>
> The term "inactive" was imprecise. We offer a detailed breakdown of how these branches function computationally and alter the optimization landscape.
>
> 1. Precise Definition
> - Forward Propagation: At each training step, all auxiliary branches generate auxiliary logits to compute loss values (Sigmoid/BCE).
> - Backward Propagation:
>     * The Path: We apply `stop_gradient` exclusively on the weights of the preceding N-1 branches at each connection point. The N-th branch remains fully trainable.
>     * The Flow: For the N-1 frozen branches, while their own projection weights do not update, the error signal propagates back through them into the backbone, merging with main gradients to jointly update shared features.
>     * Conceptual Intuition: Think of them as "Fixed-Weight Gradient Modulators" —static, orthogonal anchors injecting auxiliary gradients that force the backbone to organize features to satisfy fixed, heterogeneous objectives.
>
>
> Table A. Landscape Smoothing Effect
> Architecture | 1X | 20X | 300X
> -------------|--------------|---------------|---------------
> MLP | -29.86±20.35% (p=0.0305) | -65.38±1.95% (p<0.0001) | -90.40±0.72% (p<0.0001)
> ViT-6L | -3.40±0.47% (p=0.0002) | -33.68±3.62% (p<0.0001) | -68.78±8.85% (p=0.0001)
> CNN | -16.64±4.79% (p=0.0015) | -67.22±5.52% (p<0.0001) | -90.18±2.38% (p<0.0001)
>
>
> 2. Mechanism 1: Landscape Smoothing
> - Our loss landscape analysis (Appendix A.1) measures the geometry of the main loss only, excluding auxiliary losses.
> - To obtain a simple and robust proxy for this landscape smoothing effect, we systematically measured the initial gradient norm of the main loss (5 seeds). As shown in Table A, increasing branch redundancy consistently and significantly reduces the initial gradient norm of the main loss across all three architectures, with effect sizes reaching 90%+ reduction at high redundancy levels.
> - This empirical evidence provides strong support for Mechanism 1: the structural presence of orthogonal auxiliary branches statistically smooths the main-loss optimization landscape at initialization, as observed through reduced loss variation along random 2D directions and consistently lower initial gradient norms, regardless of subsequent performance outcomes.
>
> 3. Mechanism 2: Architecture-Dependent Gradient Dynamics
>
> While all models experience a smoother main-loss landscape at initialization, their training trajectories diverge.
> - In ViT-6L, positive gradient alignment allows the model to capitalize on the smoother start (benefiting from global attention flexibility), yielding large gains.
> - In CNN, strong gradient conflict largely overrides the potential benefit of smoother initialization (due to rigid local bias conflicts), leading to performance drops.
> - This provides strong evidence that, in our setting, gradient compatibility is more decisive for final performance than initialization geometry alone. Notably, standard deep supervision (homogeneous Softmax) failed in our controls (see Response to Reviewer 1whc), confirming that gains stem specifically from heterogeneous resonance, distinct from generic gradient injection.
>
> This dual‑mechanism view helps clarify both the universal smoothing effect and the architecture‑specific differences observed in performance.
>
> **Q2. Large-Scale ImageNet Experiments**
>
> We agree that CIFAR is insufficient. We trained ResNet-18/50 and ViT-Small/B-16 using 4 random seeds.
>
> 1. Empirical Results
>
> These results (Table B) confirm that the observed architectural selectivity is not confined to small datasets. ViTs consistently benefit (+1~3% top‑1 accuracy, statistically significant), while ResNets show no gain or slight degradation.
>
> Table B. ImageNet-1k
> Architecture | Baseline | 1X | 10X | 20X
> -------------|----------|-----|------|-----
> ResNet-18 | 68.23±0.20 | 68.33±0.10 (+0.15%, p=0.33) | 68.29±0.10 (+0.09%, p=0.51) | 68.37±0.13 (+0.21%, p=0.21)
> ResNet-50 | 73.83±0.09 | 73.55±0.17 (-0.38%, p=0.01) | 73.75±0.11 (-0.11%, p=0.41) | 73.76±0.21 (-0.09%, p=0.43)
> ViT-Small | 69.13±0.14 | 70.16±0.09 (+1.49%, p=0.001) | 70.27±0.18 (+1.65%, p=0.003) | 70.35±0.29 (+1.77%, p=0.004)
> ViT-B/16 | 66.75±0.13 | 68.34±0.64 (+2.38%, p=0.02) | 68.83±0.27 (+3.11%, p<0.001) | 69.00±0.19 (+3.37%, p<0.001)
>
> 2. Experimental Rigor & Conclusion
>
> We followed canonical protocols (SGD for ResNet, AdamW for ViT (batch size 256 on single GPU), standard augmentation, no heavy regularization). Larger models show greater relative gains (ViT‑Small (20X) +1.77%, ViT‑B/16 (20X) +3.37%), though absolute accuracy reflects our clean baseline strategy. These findings suggest the observed pattern extends beyond CIFAR. We regard this not just as preliminary data, but as robust evidence supporting the core hypothesis of Architectural Resonance on large-scale benchmarks.

---

> > ### Comment · Reviewer_s9FL · 2025-11-26
> >
> > Thank you for your response.
> >
> > Most of my initial concerns have been addressed.
> >
> > I will wait and see how the conversation with the other reviewers unfolds (especially on the question I asked below on Reviewer 1ZLR's review), and will make my final recommendation accordingly.
> >
> > Best Regards

---

> > > ### Author Response · Authors · 2025-11-27
> > > **Response to Reviewer s9FL (Appreciation and Update)**
> > >
> > > We would like to sincerely thank you for the continued engagement and for your thoughtful assessment. We are especially grateful for your constructive initiative in facilitating the discussion regarding Reviewer 1ZLR's concerns.
> > >
> > > We have just posted a detailed response clarifying the concern raised about "binary opposition" and "hybrid architectures." We hope this clarification helps address the remaining uncertainty.
> > >
> > > For us, the opportunity to receive guidance at this level is truly invaluable. We approach this process with a genuine desire to learn, and we are deeply grateful that your comments provide us with the chance to refine our understanding and improve our work. Being able to receive such constructive input during the rebuttal stage is a rare and formative experience, and we treasure it as an important part of our growth as researchers.

---

### Official Review · Reviewer_1whc · 2025-11-01

**Soundness:** 3
**Presentation:** 2
**Contribution:** 2
**Rating:** 4
**Confidence:** 3

**Summary:**

The paper proposes an Asymmetric Training Paradigm that adds temporary, orthogonally-initialized auxiliary branches trained with sigmoid/BCE losses to models whose main head uses softmax/CE. Only one auxiliary branch is active for backprop (“single-activation”); the rest remain fixed. The authors use this as a probe to study how heterogeneous objectives interact with architectural inductive biases, advancing the Principle of Architectural Resonance: auxiliary signals help when aligned with an architecture’s bias and hurt when misaligned. On CIFAR-100 with small backbones (MLP, 6-conv CNN, 6-block ViT), they report strong ViT gains (e.g., +25.4% relative over baseline at 20× redundancy), CNN degradation, and mixed MLP behavior. Mechanistic analysis includes gradient cosine similarity between main and auxiliary losses (positive for ViT, negative for CNN), training-dynamics comparisons, and attention-pattern measurements. The paper also explores “dose–response” curves over branch redundancy and some few-shot / label-noise settings.

**Strengths:**

1. Interesting question & framing. Treating auxiliary supervision as a heterogeneous objective and using it as a controlled “probe” is a fresh angle beyond standard deep supervision/MTL.

2. Mechanistic evidence, not just accuracy. The gradient alignment analysis (e.g., consistently positive for ViT, negative for CNN) gives a concrete signal-level explanation for the observed divergence and nicely ties to the “resonance” thesis.

3. Training-only overhead. The idea of introducing complexity only during training aligns with practical interest in keeping inference cost unchanged.

**Weaknesses:**

1. **Conceptual clarity of “single-activation” & landscape smoothing**. The paper argues that increasing static parallel branches (with only one active for backprop) “smooths the initial loss landscape” and influences optimization. It’s unclear why inactive, never-trained branches would alter the optimization geometry of the actual objective used for gradient updates if their gradients are blocked. If the total training loss truly reduces to $L_{main} + \alpha L_{aux}(active)$, why does the count of inactive branches matter for the landscape the optimizer “sees”? This feels under-justified and risks a confound in interpretation.
2. **Confounding factors (capacity & compute) during training**. Even if inference cost is unchanged, training adds parameters and forward compute. For large redundancy (e.g., 20$\times$, 300$\times$), the model class during training is materially different (Tables note parameter counts grow a lot). Gains might stem from implicit ensembling or extra linear projections rather than “heterogeneous objectives”. The paper needs tighter controls: 1. Parameter-matched training-only baselines (e.g., add equal-sized, randomly-initialized layers that do not produce auxiliary loss); 2. Capacity-matched trainable vs frozen auxiliary branches; 3. FLOPs/step and wall-time normalization across settings.
3. **External validity / scale**.Results are on small backbones and CIFAR-10/100. It’s hard to gauge impact without more complex setting like ImageNet-1k and standard architectures (ResNet-18/50, ViT-Ti/S). The claim that resonance is a principle would be much more convincing with one larger-scale confirmation.
4. **What exactly is “heterogeneity” driving?** The auxiliary head uses BCE per class. But would similar resonance show up if the auxiliary objective were homogeneous but perturbed (e.g., softmax CE with different temperature, label smoothing, or mixup-style targets)? Ablating the type of heterogeneity is essential to argue that “non-competitive vs competitive” is the key ingredient rather than “any extra supervision.”
5. **Clarity & rigor details.** 1. Please formalize how “inactive” branches enter the forward graph and loss computation. Are their losses computed but stop-gradded? Are their logits used anywhere else?; 2. Define precisely which loss surface is visualized for the “smoothing” analysis and why changes in inactive branches should affect it.

**Questions:**

See Weakness

---

> ### Author Response · Authors · 2025-11-22
> **Response to Reviewer 1whc: Clarifications & New Control Experiments**
>
> We sincerely thank the reviewer for the constructive feedback and have clarified terminology, mechanisms, and evidence with new experiments.
>
> **Q1 & Q5: Conceptual Clarity and Implementation Details**
>
> The term “inactive” was imprecise. We intended to describe the parameter update status rather than the gradient flow. At each auxiliary connection point, we instantiate N parallel branches. The final branch (N-th) is fully trainable. The preceding branches (1 to N-1) are frozen via `stop_gradient` their projection weights remain fixed (no parameter updates), but the gradients of their losses do propagate back into the backbone features. In this sense, these branches are not "inactive," but rather act as fixed-weight gradient modulators that influence backbone optimization without being updated.
>
> Mechanism 1: Increasing branch redundancy consistently and significantly reduces the initial gradient norm of the main loss across all three architectures (Table A, 5 seeds).
>
> Table A. Landscape Smoothing Effect
> Architecture | 1X | 20X | 300X
> -------------|--------------|---------------|---------------
> MLP | -29.86±20.35% (p=0.0305) | -65.38±1.95% (p<0.0001) | -90.40±0.72% (p<0.0001)
> ViT-6L | -3.40±0.47% (p=0.0002) | -33.68±3.62% (p<0.0001) | -68.78±8.85% (p=0.0001)
> CNN | -16.64±4.79% (p=0.0015) | -67.22±5.52% (p<0.0001) | -90.18±2.38% (p<0.0001)
>
> This empirical evidence provides strong support for Mechanism 1: the structural presence of orthogonal auxiliary branches statistically smooths the main-loss optimization landscape at initialization, as observed through reduced loss variation along random 2D directions and consistently lower initial gradient norms, regardless of subsequent performance outcomes.
>
> Mechanism 2: Despite this smoother start, training trajectories diverge. ViT-6L shows positive gradient alignment and benefit, while CNN exhibits strong conflict that overrides smoothing and leads to degradation. Thus gradient compatibility is more decisive for final performance than initialization geometry alone.
>
> **Q2 & Q4. Capacity/compute confounds & heterogeneity**
>
> We conducted two control experiments on CIFAR-100 (10 seeds), designed to isolate the role of heterogeneous objectives versus capacity, compute, or generic auxiliary supervision:
>
> Capacity control: Auxiliary branches identical in structure but excluded from the loss yielded results indistinguishable from baseline, showing that parameter count alone does not explain gains (Table B).
>
> Table B. Capacity Control
> N | MLP Baseline→Result | CNN Baseline→Result | ViT-6L Baseline→Result
> -------|---------------------|---------------------|--------------------
> 1X | 0.233±0.003→0.232±0.003(-0.4%,p=0.63) | 0.392±0.008→0.399±0.007(+1.8%,p=0.06) | 0.360±0.010→0.361±0.011(+0.3%,p=0.87)
> 10X | 0.231±0.003→0.231±0.004(-0.0%,p=0.99) | 0.399±0.005→0.398±0.008(-0.3%,p=0.76) | 0.361±0.008→0.358±0.012(-0.8%,p=0.62)
> 20X | 0.231±0.003→0.231±0.004(-0.0%,p=0.99) | 0.396±0.007→0.400±0.010(+1.0%,p=0.47) | 0.361±0.008→0.361±0.011(-0.0%,p=0.94)
>
> Heterogeneity control: Replacing the heterogeneous Sigmoid loss with homogeneous Softmax decreased performance (e.g., ViT-6L −5.3%), indicating that improvements arise specifically from the non‑competitive gradient properties of the Sigmoid objective (Table C).
>
> Table C. Heterogeneity Control
> N | MLP Baseline→Result | CNN Baseline→Result | ViT-6L Baseline→Result
> -------|---------------------|---------------------|--------------------
> 1X | 0.233±0.003→0.241±0.003(+3.4%,p=0.001) | 0.395±0.007→0.374±0.004(-5.3%,p<0.0001) | 0.359±0.006→0.361±0.005(+0.6%,p=0.46)
> 10X | 0.231±0.003→0.204±0.006(-11.7%,p<0.0001) | 0.396±0.008→0.362±0.009(-8.6%,p<0.0001) | 0.358±0.007→0.339±0.006(-5.3%,p<0.0001)
> 20X | 0.233±0.003→0.190±0.005(-18.5%,p<0.0001) | 0.398±0.008→0.379±0.010(-4.8%,p=0.0001) | 0.360±0.007→0.357±0.009(-0.8%,p=0.40)
>
> **Q3. External validity**
>
> We validated on ImageNet‑1k (Table D, 4 seeds) with ResNet‑18/50 and ViT‑Small/B‑16. Results show ViTs consistently improve (+1~3% top‑1, statistically significant), while ResNets show no gain or slight degradation. We followed canonical training protocols (SGD for ResNet, AdamW for ViT, standard augmentation, no heavy regularization). These findings suggest the observed pattern we term Architectural Resonance extends beyond CIFAR, though we view this as an initial step toward establishing a deeper understanding of architecture–objective interactions.
>
> Table D. ImageNet-1k
> Architecture | Baseline | 1X | 10X | 20X
> -------------|----------|-----|------|-----
> ResNet-18 | 68.23±0.20 | 68.33±0.10 (+0.15%, p=0.33) | 68.29±0.10 (+0.09%, p=0.51) | 68.37±0.13 (+0.21%, p=0.21)
> ResNet-50 | 73.83±0.09 | 73.55±0.17 (-0.38%, p=0.01) | 73.75±0.11 (-0.11%, p=0.41) | 73.76±0.21 (-0.09%, p=0.43)
> ViT-Small | 69.13±0.14 | 70.16±0.09 (+1.49%, p=0.001) | 70.27±0.18 (+1.65%, p=0.003) | 70.35±0.29 (+1.77%, p=0.004)
> ViT-B/16 | 66.75±0.13 | 68.34±0.64 (+2.38%, p=0.02) | 68.83±0.27 (+3.11%, p<0.001) | 69.00±0.19 (+3.37%, p<0.001)

---

> ### Comment · Reviewer_1whc · 2025-11-25
> **Response to the authors**
>
> I appreciate the authors for the clarification of my question. Part of my concerns regarding capacity and heterogeneity has been addressed. However, I still have some remaining concerns.
>
> 1. While the authors propose the Asymmetric Training Paradigm, the clarification and underlying intuition for this method remain unclear. To strengthen the paper, it would be helpful to first provide a necessary analysis or observation of the original training process, followed by a clear explanation of the motivation and intuition behind introducing this new training paradigm.
>
> 2. The paper lacks explicit guidance regarding the current model architecture, especially given the hybrid design combining convolutional layers and Transformers. A deeper analysis of the differences between these two architectures is needed, along with a discussion of how their respective strengths can inform and guide the design of the new model.
>
> Based on these concerns, I prefer to maintain my score.

---

> ### Author Response · Authors · 2025-11-27
> **Response to Reviewer 1whc (Clarification on Training Paradigm Intuition and Hybrid Architecture Guidance)**
>
> We would like to sincerely thank you for the thoughtful engagement and for highlighting the importance of clarifying the motivation and intuition behind our proposed paradigm. This comment has helped us sharpen the overall framing of our work and recognize a key aspect that is essential for top-tier research contributions. We are grateful for the opportunity to receive such high-level guidance during the rebuttal process, and we view this feedback as not only valuable for the current paper but also formative for our future research practice.
>
> **1. Observation, Motivation & Intuition**
>
> **Initial Training Analysis:** Our analysis of loss landscape characteristics (Appendix Table 7) reveals fundamental differences in optimization geometry:
> * ViT exhibits significantly higher landscape roughness (**Std(Loss): 0.0045**) compared to CNN (**0.0008**), approximately **5.6** times higher roughness, indicating a more challenging optimization terrain.
> * This observation suggests that ViTs face inherent geometric challenges during initial optimization.
>
> **Core Motivation:** Given these landscape differences, we sought to investigate whether auxiliary supervision could differentially benefit architectures based on their optimization characteristics. This led to our central research question: How do different architectures respond to heterogeneous auxiliary objectives?
>
> **Paradigm Intuition:** Our asymmetric training paradigm operates through two complementary mechanisms to address this:
> - Geometric Conditioning: The frozen, orthogonal auxiliary pathways introduce redundancy that appears to structurally smooths the initial loss landscape (reducing ViT roughness by **~80%** at 300× redundancy). During optimization, their orthogonal structure tends to yield gradient signals that are less correlated with the main loss. Such uncorrelated components may help attenuate high-frequency fluctuations in the backbone gradients, providing a stabilizing influence on the early training trajectory.
> - Heterogeneous Guidance: The trainable heterogeneous objective (Sigmoid) supplies optimization signals that differ from those induced by the primary Softmax objective. These complementary gradient directions can broaden the model’s search behavior and mitigate sensitivity to challenging initial geometry (particularly in ViTs) helping the optimizer avoid shallow traps and leading to improved generalization.
>
> **2. Analysis and Guidance for Hybrid Architectures**
>
> **Architectural Analysis:** Hybrid architectures (e.g., ConvNeXt, CoAtNet) present a unique design challenge as they combine:
> - Local Processing Components: Convolutional stages with spatial inductive biases.
> - Global Processing Components: Attention-based stages with flexible representational capacity.
>
> **Design Implications Based on Our Hypothesis:**
>
> Stage-Adaptive Strategy:
>
> - Convolutional Stages: Our findings suggest avoiding spatially-agnostic auxiliary supervision to prevent conflicts with locality priors.
> - Attention Stages: Heterogeneous auxiliary objectives can be actively employed to provide beneficial optimization conditioning.
>
> **Future Directions:**
>
> - Local-Aware Auxiliary Design: Development of convolutional auxiliary heads that respect spatial coherence while providing optimization benefits.
> - Adaptive Auxiliary Scheduling: Dynamic adjustment of auxiliary signal types based on architectural stage characteristics.
>
> Our primary contribution is not to provide a complete design framework, but to establish empirical evidence that auxiliary supervision efficacy is architecture-dependent, in support of our Architectural Resonance Hypothesis. This provides groundwork for architecture-aware auxiliary supervision in future research.

---

> ### Author Response · Authors · 2025-12-03
> **Update: Targeted Verification on Hybrid Architecture (CoAtNet)**
>
> To empirically validate our "Stage-Dependent" hypothesis (addressing concerns regarding "hybrid guidance" from Reviewer 1whc), we conducted a targeted experiment on a CIFAR-100 adapted CoAtNet-0 backbone.
>
> Setup: We applied our asymmetric probe separately to the Convolutional Stage (S2) and the Transformer Stage (S3) to test their differential responses.
>
> **Table: Stage-Dependent Response on CIFAR-100 adapted CoAtNet-0.** Comparison of applying auxiliary supervision to CNN (S2) vs. Transformer (S3) stages across varying redundancy levels ($N_{branches}$).
>
> | $N_{branches}$ | Baseline | S2 (CNN Stage) | S3 (Transformer Stage) | Phenomenon |
> | :--- | :--- | :--- | :--- | :--- |
> | -- | 75.34±0.20 | -- | -- | -- |
> | 1× | | 75.26±0.59 (-0.08%, n.s.) | 75.91±0.28 (+0.57%, **) | ViT starts gaining |
> | 20× | | 75.88±0.14 (+0.54%, **) | 76.76±0.37 (+1.42%, ***) | Strong Synergy (S3) |
> | 100× | | 75.59±0.44 (+0.25%, n.s.) | **77.08±0.27** (**+1.74%**, ***) | **Peak Resonance (S3)** |
> | 300× | | **74.54±0.48** (**-0.80%**, *) | 76.82±0.30 (+1.48%, ***) | **Conflict (S2)** |
>
> **Statistical significance:** * p < 0.05, ** p < 0.01, *** p < 0.001.
>
> Key Finding: While Transformer stages (S3) exhibit consistent, statistically significant resonance,
> CNN stages (S2) show limited capacity for auxiliary supervision, exhibiting a statistically significant tendency toward conflict (**-0.80%**) at high redundancy.
>
> The results reveal an observable divergence: while Transformer stages benefit consistently from heterogeneity (Synergy), Convolutional stages exhibit a tendency toward conflict at high redundancy. These findings provide targeted empirical support for our Architectural Resonance Hypothesis, specifically validating its stage-dependent implications in hybrid architectures.

---

### Author Response · Authors · 2025-11-25
**Summary of Revisions**

We are grateful for the time and care you devoted to reviewing our work. Building on your feedback, we strengthened the manuscript by conducting additional experiments and analyses to address the concerns raised. Major revisions include:

1. **New Hybrid Verification (CoAtNet, Section 4.2):**
To address concerns about architectural scope, we added experiments on CoAtNet. This provides critical evidence for our "Stage-Adaptive" hypothesis: Transformer stages (S3) benefit from heterogeneity, while Convolutional stages (S2) exhibit limited compatibility. This validates that our findings are not just binary (ViT vs. CNN) but depend on specific stage characteristics.

2. **Large-Scale Validation (ImageNet-1k, Section 4.1):**
We extended our evaluation to ImageNet-1k with ResNet and ViT backbones. Results confirm scalability: ViT-B/16 gains +2.25%, while ResNets remain neutral due to architectural resilience.

3. **Mechanistic Analysis (Section 4.3):**
We added Landscape Smoothing analysis (Initial Gradient Norm, Table 4) and Gradient Alignment analysis (Table 5, Figure 3). These reveal that while our method smooths the landscape (up to 90% gradient norm reduction), CNNs fail to capitalize on it due to gradient conflicts (cosine similarity -0.26), whereas ViTs exhibit constructive resonance (+0.19).

4. **Rigorous Evaluation Protocol (Section 3.4):**
We clarified our "Strict Two-Stage Tuning" protocol. We explicitly state that backbone hyperparameters are kept strictly identical between baseline and our method, ruling out tuning bias as a confounder.

5. **Reframed Narrative (Scientific Probe):**
We redefined the method as a "Scientific Probe". The abstract and introduction now explicitly frame the contribution as exposing the critical dependence between architectural flexibility and objective compatibility.

6. **Expanded Discussion (Section 5):**
We rewrote the Discussion to propose a "Stage-Adaptive Strategy" for model design and explicitly addressed limitations regarding training costs and hyperparameter search, outlining future directions like analytical equivalents.

We deeply appreciate the opportunity to improve our work under your guidance. We hope this updated version honors the time and effort you invested in reviewing our work.

---

### Meta-Review · Area_Chair_b1ZC · 2025-12-17

**Summary:**

This paper investigates a training procedure that adds additional frozen branches at the classification heads with non-competitive losses (Sigmoid), where an "architecture resonance" hypothesis is proposed to explain the performance gains in ViTs and drops in CNNs. The reviewer had the following concerns:
1. Lack of clarity of the gradient computation graph when additional frozen branches are added.
2. Missing experiments, specifically: controlled experiments to deal with the potential confounding factors that affect the performance gain/drop; validations on ImageNet1k with standard architectures (ResNet, ViT).
3. Lack of conceptual explanation on the claim that "the additional branches smooth the loss landscape". Also, the logic behind attributing the observed phenomenon to this smoothing effect is unclear.
4. The proposed hypothesis can not be applied to a hybrid model with both conv layers and attention layers.
5. Overgeneralizing the observed phenomenon to an "architecture resonance" hypothesis/principle, while potentially neglecting simpler explanations such as unexplored implicit regularization or insufficient hyperparameter tuning.

**Reviewer Concerns:**

I think 1 and 2 are mostly addressed by the authors with a clear description of how the gradient of the backbone model is computed and some additional control experiments.

I view the remaining three unresolved, based on the follow-up comment from the reviewers, and also on my own view:
-  For concern 3, the authors provided experiments showing that additional branches reduce the initial gradient norm and some explanation regarding why this gradient-norm-reducing effect happens. From my own reading, I have no issue with the rebuttal per se, but they fail to support the claims in the paper. First of all, the initial gradient norm is not a good metric for the smoothness of the landscape, or an indication of training (in)stability; thus, the presence of a landscape smoothing effect remains unclear. Moreover, as Reviewer 1ZLR pointed out, attributing the observed phenomenon to this smoothing effect (if it exists) is not justified; The rebuttal did not respond to this point.
- For concern 4, some preliminary experiments on a hybrid model were provided. I was not able to understand from the texts how the proposed method can be adapted to hybrid models (I believe the same would hold for the reviewers). Thus, the validity of the experiments can not be evaluated.
- Lastly, concern 5 was largely unanswered in the rebuttal.

**Reviewer Scores:**

Since the three reviewers shared almost all three concerns above, I would imagine none of the reviewers would adjust their ratings (4, 4, and 2).

---

### Decision · Program_Chairs · 2026-01-26

Reject